# Cholesterol ensures ciliary polycystin-2 localization to prevent polycystic kidney disease

Takeshi Itabashi[1,2,*], Kosuke Hosoba[3,4,*], Tomoka Morita[1,2,*], Sotai Kimura[5,6], Kenji Yamaoka[7], Moe Hirosawa[1,2], Daigo Kobayashi[1], Hiroko Kishi[1,8], Kodai Kume[9], Hiroshi Itoh[5], Hideshi Kawakami[9], Kouichi Hashimoto[7], Takashi Yamamoto[3,4], Tatsuo Miyamoto[1,2]

**The plasma membrane covering the primary cilium has a diverse accumulation of receptors and channels. To ensure the sensor function of the cilia, the ciliary membrane has higher cholesterol content than other cell membrane regions. A peroxisomal biogenesis disorder, Zellweger syndrome, characterized by polycystic kidney, is associated with a reduced level of ciliary cholesterol in cells. However, the etiological mechanism by which ciliary cholesterol lowering causes polycystic kidney disease remains unclear. Here, we demonstrated that lowering ciliary cholesterol by either pharmacological treatment or genetic depletion of peroxisomes impairs the localization of a ciliary ion channel polycystin-2. We also generated cultured renal medullary cells and mice carrying a missense variant in the cholesterol-binding site of polycystin-2 detected in the patient database of autosomal dominant polycystic kidney disease. This missense protein showed normal channel activity but decreased localization to the ciliary membrane. The homozygous mice exhibited embryonic lethality and the ciliopathy spectrum conditions of situs inversus and polycystic kidney. Our results suggest that cholesterol controls the ciliary localization of polycystin-2 to prevent polycystic kidney disease.**

## Introduction

Primary cilia are nonmotile, antenna-like organelles that develop on the surface of most mammalian cells. The cell membrane surrounding the axonemal microtubules polymerized from the centrosome-converted basal body is known as the ciliary membrane, which contains a variety of receptors and ion channels, thereby sensing extracellular chemical and physical stimuli to transduce signals (Nachury & Mick, 2019; Derderian et al, 2023). Defects in primary cilia cause ciliopathies characterized by a range of clinical symptoms, including polycystic kidney, polydactyly, retinitis pigmentosa, and neuronal and other developmental abnormalities (Baker & Beales, 2009). To date, at least 38 ciliopathies with mutations in more than 240 genes have been identified (Reiter & Leroux, 2017). Ciliopathies have been classified into "first-order" ciliopathies caused by mutations in genes encoding primary cilia and centrosome proteins and "second-order" ciliopathies associated with mutations in cilium- and centrosome-unrelated genes (Lovera & Luders, 2021). Uncovering the genes underlying these second-order ciliopathies might shed light on how different organelles act as unexpected regulators of primary cilia.

Among the second-order ciliopathies, Zellweger syndrome (ZS) is an autosomal recessive peroxisome biogenesis disorder caused by germline mutations of the peroxisomal biogenesis factor (PEX) genes, which encode proteins called peroxins (Gould & Valle, 2000; Fujiki et al, 2022). Patients with ZS show severe neonatal hypotonia and liver dysfunction because of the absence or reduction of functional peroxisomes, which are involved in α- and β-oxidation of fatty acids, amino acid synthesis, and metabolism of reactive oxygen species (ROS) and bile acids (Argyriou et al, 2016; Waterham et al, 2016). They also often exhibit ciliopathy-like symptoms such as polycystic kidney and retinitis pigmentosa (Luisiri et al, 1988; Folz & Trobe, 1991; FitzPatrick, 1996; Klouwer et al, 2015). ZS is thus known as cerebrohepatorenal syndrome. Although the clinical manifestations of ZS in the brain and liver might be attributable to defects in a typical function of peroxisomes, such as lipid metabolism, it is unclear how the peroxisomal failure causes the cilium-related polycystic kidney. Previously, we demonstrated that peroxisomes

[1]Department of Molecular and Cellular Physiology, Graduate School of Medicine, Yamaguchi University, Yamaguchi, Japan   [2]Division of Advanced Genome Editing Therapy, Research Institute for Cell Design Medical Science, Yamaguchi University, Yamaguchi, Japan   [3]Program of Biomedical Science, Graduate School of Integrated Sciences for Life, Hiroshima University, Hiroshima, Japan   [4]Program of Mathematical and Life Science, Graduate School of Integrated Sciences for Life, Hiroshima University, Hiroshima, Japan   [5]Department of Molecular Pathology, Graduate School of Medicine, Yamaguchi University, Yamaguchi, Japan   [6]Department of Anatomic Pathology, Hirosaki University Hospital, Aomori, Japan   [7]Department of Neurophysiology, Graduate School of Biomedical and Health Sciences, Hiroshima University, Hiroshima, Japan   [8]Department of Environmental Physiology, Faculty of Medicine, Shimane University, Shimane, Japan   [9]Department of Molecular Epidemiology, Research Institute for Radiation Biology and Medicine, Hiroshima University, Hiroshima, Japan

Correspondence: t-miyamoto@yamaguchi-u.ac.jp
*Takeshi Itabashi, Kosuke Hosoba, and Tomoka Morita contributed equally to this work

move along the microtubules toward primary cilia, thereby trafficking cholesterol to the ciliary membrane (Miyamoto et al, 2020). However, the etiological mechanisms of polycystic kidney observed in ZS remain unclear. Against this background, the ZS-related pathological link between polycystic kidney and ciliary cholesterol insufficiency led us to explore how cholesterol controls the physiological roles of primary cilia in kidney epithelia.

Polycystin-1 (PC1) and polycystin-2 (PC2) form a heteromeric cation channel complex (hereafter called the polycystin complex) at a ratio of 1:3 localized in the primary cilium membrane of kidney epithelial cells (Qian et al, 1997; Tsiokas et al, 1997; Hanaoka et al, 2000; Yu et al, 2009; Su et al, 2018). It is widely considered that the polycystin complex senses extracellular stimuli such as mechanical sheer stress or chemical ligands to control renal epithelial cell proliferation and differentiation (Douguet et al, 2019). PC1, encoded by the *PKD1* gene, is an eleven-transmembrane protein with a large extracellular N-terminal fragment (Malhas et al, 2002). PC2 encoded by the *PKD2* gene is a six-transmembrane protein with nonselective cation ion channel activity belonging to the transient receptor potential polycystic (TRPP) ion channel family. It has been proposed that PC2 could play roles in extracellular sheer stress-dependent intracellular $Ca^{2+}$ signaling in the primary cilia of the renal epithelium (Nauli et al, 2003) and the intracellular $Ca^{2+}$ release channel in endoplasmic reticulum (Koulen et al, 2002). In contrast, recent studies have suggested that PC2 in the ciliary membrane is a nonselective cation channel, with a low level of $Ca^{2+}$ current, rather than a $Ca^{2+}$ channel (Delling et al, 2016). Germline mutations in *PKD1* and *PKD2* genes cause autosomal dominant polycystic kidney disease (ADPKD), characterized by the enlargement of renal tubules and the formation of isolated fluid-filled cysts within the kidney and other organs such as the liver and pancreas. Most cases of ADPKD (70%) are caused by mutations in the *PKD1* gene, whereas *PKD2* gene mutations are responsible for ~15% of cases. It is estimated that ADPKD affects 1 in 1,000 people and slowly progresses to end-stage renal disease in about half of the individuals with *PKD1* or *PKD2* heterozygous mutation (Wilson, 2004). It is thought that polycysts in the patients occur via a cellular recessive two-hit mechanism of germline and somatic mutations in the *PKD1* or *PKD2* gene (Saigusa & Bell, 2015).

Cholesterol is known as a modulator of ion channels and chemical receptors. A combined analysis involving cryo-electron microscopy and molecular dynamics simulations revealed a cholesterol-binding site on the outer-leaflet-facing surface of the PC2 molecule (Wang et al, 2020). The steroid nucleus of cholesterol fits within a hydrophobic pocket formed by a group of leucine (Leu517 and Leu656), isoleucine (Ile561 and Ile659), and valine (Val564 and Val655) residues located between the S3 and S4 helices of the voltage sensor–like domain (VSLD) and pore domain S6 helix. Notably, a missense variant of Leu517Arg forming the cholesterol-binding site of PC2 is included in the genetic database of ADPKD (see http://pkdb.pkdcure.org) and categorized as "likely pathogenic," suggesting that cholesterol might be required for the polycystin complex activity. However, no reverse genetics proof that defects in the interaction between cholesterol and PC2 cause polycystic kidney diseases has yet been obtained.

In this study, we demonstrate that *Pex14*-deficient mouse collecting duct–derived (mIMCD3) cells as a ZS model show reduced levels of cholesterol and PC2 in the ciliary membranes and that cholesterol administration restores the ciliary localization of PC2 and the epithelial lumen architecture in these cells. Using CRISPR/Cas9 technology, we generated PC2 cholesterol-binding-site mutant mIMCD3 cell lines and mice. We also found that the physical interaction between cholesterol and PC2 is required for PC2 ciliary localization to prevent polycystic kidney disease.

# Results

## Localization of polycystin requires peroxisome-derived cholesterol in primary cilia

We previously reported that ZS is associated with insufficient ciliary cholesterol levels, which might cause polycystic kidney (Miyamoto et al, 2020). First, we examined the localization of the polycystin complex components, PC1 and PC2, in the cholesterol-reduced primary cilium. Immunofluorescence using confocal microscopy was performed on mIMCD3 cells using antibodies against polycystins, ARL13B for primary cilia and FGFR1OP or γ-tubulin for the basal body. Measurement of the fluorescence intensity of fluorescently labeled antibodies and a cholesterol probe, filipin III, showed that the ciliary accumulation of both PC1 and PC2 was inhibited (62% of the level of untreated WT cells for PC1 and 50% for PC2) (Fig 1A–C) by methyl-β-cyclodextrin (MβCD) treatment, which caused a decrease in ciliary cholesterol (Fig 1D and E). Live-cell imaging for mPkd2-*Aequorea coerulescens* green fluorescent protein (AcGFP) transiently expressed in *Pkd2*-knockout mIMCD3 cells showed that MβCD treatment reduced the ciliary signal level of the PC2-AcGFP fusion protein without any remarkable endocytosis-like PC2 vesicles around the basal body (Pedersen et al, 2016) (Fig S1A and B), implying that preexisting ciliary PC2 proteins are internalized through lateral diffusion. Because peroxisomes are essential for the transport of cholesterol into the ciliary membrane (Miyamoto et al, 2020), we generated *Pex14*-knockout mIMCD3 cells as a Zellweger syndrome model using the nonhomologous end-joining (NHEJ)–mediated targeting method named ObLiGaRe (obligate ligation-gated recombination) (Maresca et al, 2013; Royba et al, 2017) (Tables S1 and S2) to evaluate whether ciliary cholesterol modulates the localization of the polycystin complex. In the $Pex14^{-/-}$ mIMCD3 cell clones obtained by this method, the PEX14 protein was not detected by Western blot analysis, resulting in a reduction of ciliary cholesterol (Miyamoto et al, 2020) (Figs 1D and E and S2A). Although the $Pex14^{-/-}$ cells did not show any impairment of ciliogenesis (Fig S2B), the level of the polycystin complex was decreased in the cholesterol-reduced primary cilia (61% of the level of untreated WT cells for PC1 and 52% for PC2) (Fig 1A–C). Interestingly, however, the addition of water-soluble cholesterol, which directly inserts cholesterol into the plasma membrane to increase the amount of ciliary cholesterol (Fig 1D and E), resulted in the restoration of polycystin complex localization (105% of the level of untreated WT cells for PC1 and 100% for PC2) (Fig 1A–C).

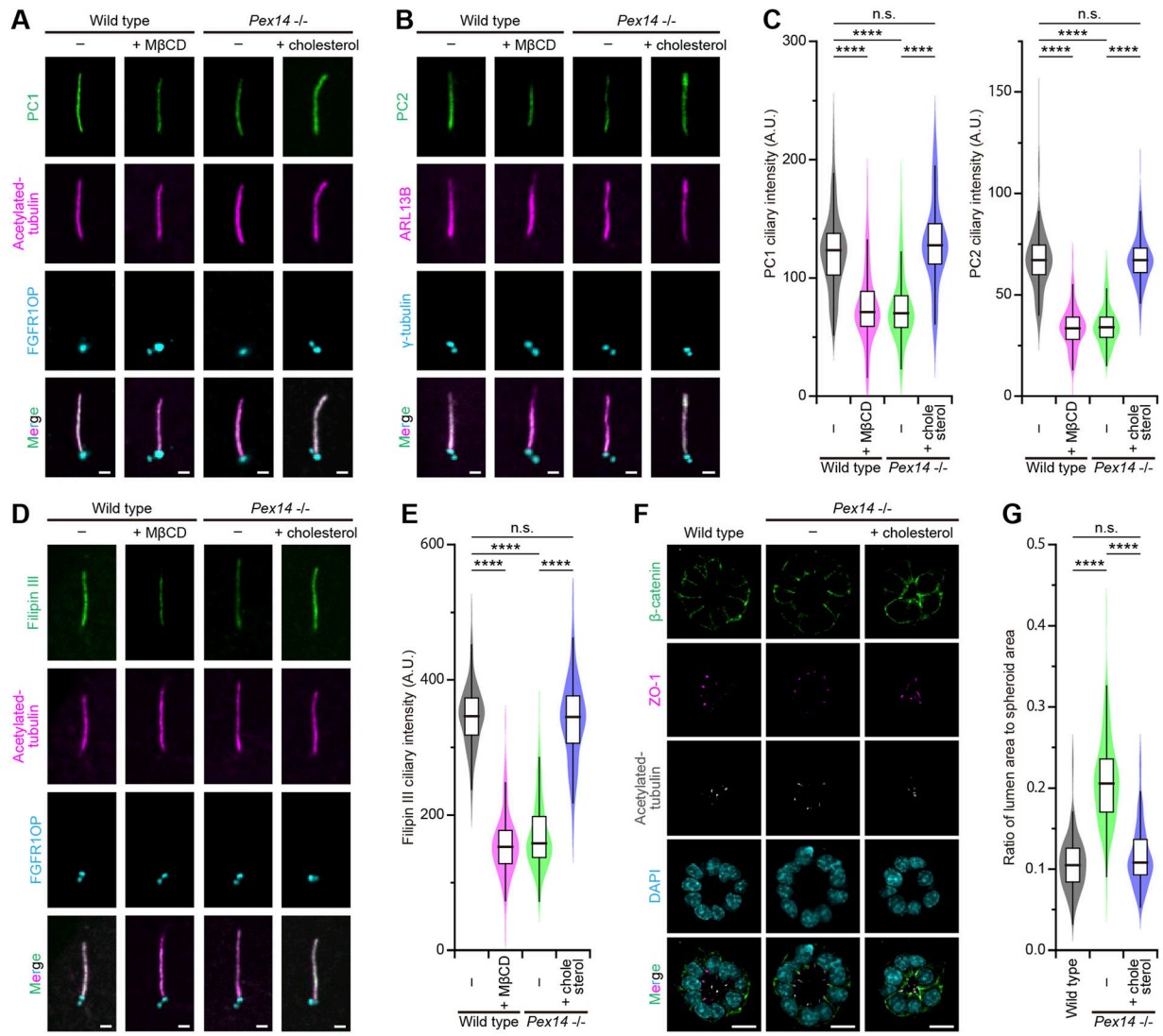

**Figure 1. Zellweger syndrome exhibits insufficient levels of ciliary cholesterol and the polycystin complex.**
**(A, B)** Ciliary localization of PC1 (A) and PC2 (B) in quiescent $G_0$-phase WT and $Pex14^{-/-}$ mIMCD3 cells. **(A, B)** WT cells were treated with methyl-$\beta$-cyclodextrin, and the $Pex14^{-/-}$ cells were incubated with cholesterol (cholesterol/methyl-$\beta$-cyclodextrin complex), and then, the cells were immunostained with anti-PC1 (green), anti-acetylated tubulin (magenta), and FGFR1OP (cyan) antibodies (A), or with anti-PC2 (green), anti-ARL13B (magenta), and anti-$\gamma$-tubulin (cyan) antibodies (B). Scale bars, 1 $\mu$m. **(C)** Quantification of the PC1 (left) and PC2 (right) intensities at primary cilia in WT cells with or without M$\beta$CD treatment and $Pex14^{-/-}$ cells with or without cholesterol treatment (****$P < 0.0001$; n.s., not significant, one-way ANOVA with Tukey's multiple comparison tests, n = 3 independent experiments: 50–54 cells per experiment). Graphs show the median and quartiles. **(D)** Localization of cholesterol in primary cilia. The WT and $Pex14^{-/-}$ cells were stained with a cholesterol probe, filipin III (green), anti-acetylated tubulin (magenta), and FGFR1OP (cyan) antibodies. Scale bars, 1 $\mu$m. **(E)** Filipin III ciliary intensity in WT cells with or without M$\beta$CD treatment and $Pex14^{-/-}$ cells with or without cholesterol treatment (****$P < 0.0001$; n.s., not significant, one-way ANOVA with Tukey's multiple comparison tests, n = 3 independent experiments: 50–53 cells per experiment). The graph shows the median and quartiles. **(F)** Spheroid formation in the WT and $Pex14^{-/-}$ mIMCD3 cells grown in three-dimensional culture. After the formation of spheroids, $Pex14^{-/-}$ cells were treated with cholesterol or left untreated. Then, the cells were stained for $\beta$-catenin (green), ZO-1 (magenta), acetylated tubulin (white), and DAPI (cyan). Scale bars, 10 $\mu$m. **(G)** Ratio of lumen area to spheroid area in WT cells and $Pex14^{-/-}$ cells with or without cholesterol treatment (****$P < 0.0001$; n.s., not significant, one-way ANOVA with Tukey's multiple comparison tests, n = 3 independent experiments: 45–61 spheroids per experiment). The graph shows the median and quartiles.

These results indicate that the localization of the polycystin complex in the primary cilia is dependent on the cholesterol derived from peroxisomes.

To pursue this finding further, we introduced a three-dimensional culture model in which mIMCD3 cells were grown using 3D Matrigel to replicate the morphology of kidney epithelial cells. mIMCD3 cell lines were grown in the 3D culture systems for

3 d to generate spheroid formation, followed by 1-d exposure to serum-free medium for ciliogenesis (Fig S2C). Then, spheroids were stained for acetylated $\alpha$-tubulin (primary cilium marker), ZO-1 (epithelial tight junction marker), and $\beta$-catenin (adherens junction marker) to confirm the apicobasal polarity (Figs 1F and S2D and E). It was reported that the lumen sizes of ciliopathy-related gene–disrupted mIMCD3 spheroids were enlarged (Luijten et al,

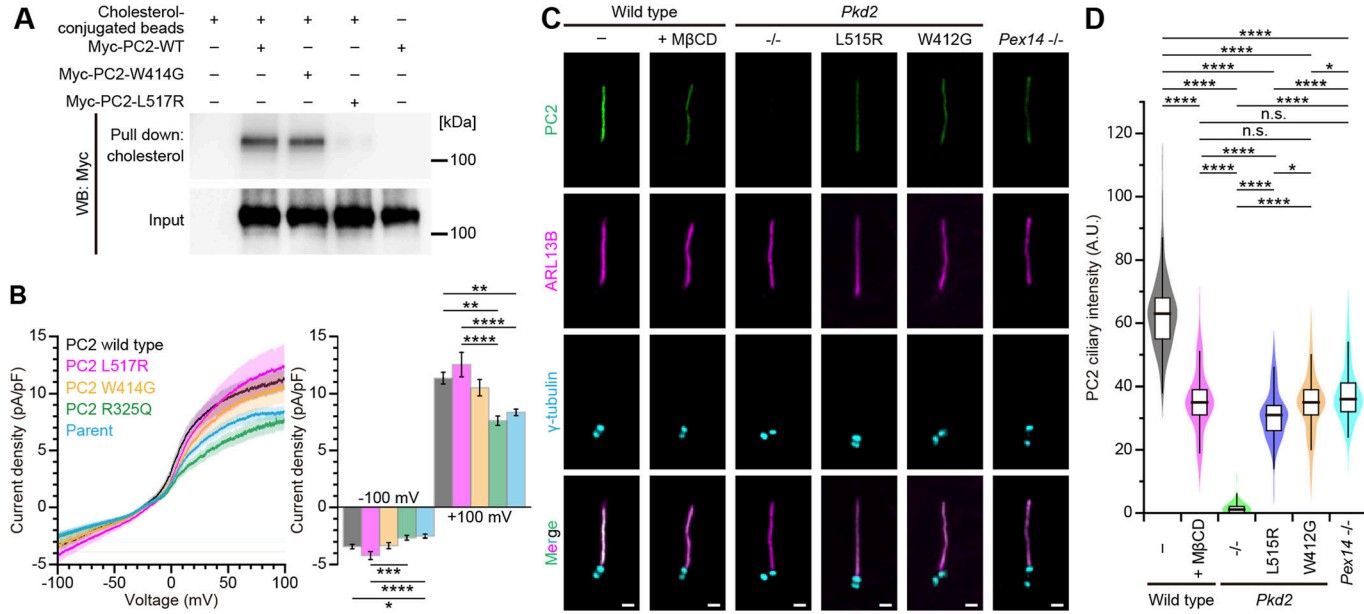

**Figure 2. Peroxisome-derived cholesterol is involved in the ciliary localization of polycystin-2.**
**(A)** Western blot analysis for pull-down assay. The lysates of HEK293T cells transiently expressing either Myc-tagged PC2 WT, PC2 W414G, or PC2 L517R mutant protein were precipitated with cholesterol-conjugated magnetic beads and analyzed by Western blotting using anti-Myc antibody. **(B)** Average current density changes in response to a voltage ramp from −100 to 100 mV in AcGFP-tagged PC2 variants. Current densities are calculated by dividing the individual current by membrane capacitance of the HEK293 cell. Current densities at −100 mV and 100 mV are shown on the right. Untransfected parent cells and PC2 R325Q–expressing cells were used as a negative control (Vien et al, 2020). Error bars represent the mean ± SEM (n.s., not significant, one-way ANOVA with Tukey's multiple comparison tests, n = 22 [WT], 21 [L517R], 14 [W414G], 14 [R325Q], and 21 [untransfected parent] cells). **(C)** Ciliary localization of PC2 variants in quiescent $G_0$-phase mIMCD3 cells. Cells were stained with anti-PC2 (green), anti-ARL13B (magenta), and anti-γ-tubulin (cyan) antibodies. Line-scan PC2 intensity of each primary cilium is shown in Fig S5D. Scale bars, 1 μm.
**(D)** Quantification of the PC2 intensity at primary cilia in *Pkd2*-mutated cells (*P < 0.05; **P < 0.01; ***P < 0.001; ****P < 0.0001; n.s., not significant, one-way ANOVA with Tukey's multiple comparison tests, n = 3 independent experiments: 50–55 cells per experiment). The graph shows the median and quartiles.

2013; Giles et al, 2014), suggesting that the ratio of lumen size to spheroid size serves as a readout of primary ciliary function in the 3D epithelial architecture. By performing 3D observation, the cross-sectional plane (xy-plane) associated with the largest size of the spheroid was determined, and then, the spheroid and lumen areas were measured in the same xy-plane (Fig S2F). In 3D cultures of untreated WT mIMCD3 cells, the ratio of lumen area to spheroid area was 0.11 ± 0.03 (mean ± SD) (Fig 1G). The addition of water-soluble cholesterol to WT cells for 8 h did not produce any changes in the epithelial lumen architecture (Fig S2D, E, G, and H). Mean-while, the treatment with MβCD for 2 h after the completion of spheroid formation induced an increase in the lumen size, causing the ratio of lumen area to spheroid area to become significantly greater (0.21 ± 0.06, mean ± SD) (Fig S2I). The spheroids composed of the *Pex14*$^{−/−}$ mIMCD3 cells also had a large lumen-to-spheroid area ratio (0.21 ± 0.05, mean ± SD), but the treatment with exogenous cholesterol improved the lumen size and this ratio to the untreated WT levels (0.12 ± 0.04, mean ± SD) (Fig 1G). These results indicate that cholesterol administration restores the epithelial lumen size in ZS model cells.

Taken together, our findings demonstrate that the polycystin complex localizes to primary cilia in a manner dependent on peroxisome-mediated ciliary cholesterol, by which the epithelial lumen architecture is regulated. This suggests that the functional polycystin complex may need to directly or indirectly interact with cholesterol.

## Cholesterol binding of polycystin-2 is required for ciliary localization

In silico analysis based on cryo-EM protein structure implied that PC2 of the polycystin complex has a cholesterol-binding site (Wang et al, 2020), Leu517Arg of which is reported to be one of the likely pathogenic missense variants of ADPKD. To ascertain whether PC2 directly binds cholesterol, we performed a pull-down assay with cholesterol-conjugated beads using the whole lysate of HEK293T cells transiently expressing either the Myc-tagged PC2 WT or the PC2 L517R mutant protein. Interaction of PC2 with cholesterol was confirmed in the PC2 WT but not in PC2 L517R (Figs 2A and S3A), demonstrating that Leu517 of PC2 is related to cholesterol binding. PC2 is a six-transmembrane protein with nonselective cation ion channel activity, so we measured the channel activity of this mutant to examine whether the impairment in the cholesterol-binding site of PC2 alters the channel function. PC2, the C terminus of which was tagged with AcGFP, was transfected into HEK293 cells to establish stably expressing cell lines (Fig S3B and C). Membrane currents in response to a 1-s voltage ramp from −100 to 100 mV were measured from AcGFP stably expressing cells using whole-cell recording. Overall membrane currents were not altered in PC2 L517R-expressing cells compared with the level in WT PC2-expressing ones (Fig 2B). The ion channel activity of the PC2 W414G mutant with the cholesterol-binding ability (Fig 2A), which remains controversial (Cai et al, 2014; Wang et al, 2023), is likely no apparent differences

between the WT and the L517R mutant (Fig 2B). For comparison, we performed a similar recording from HEK293 cells expressing the PC2 R325Q mutant, the ion channel activity of which was reported to be impaired (Vien et al, 2020). Overall membrane currents were significantly decreased in PC2 R325Q–expressing and untransfected parent cells compared with the level in WT PC2–expressing and PC2 L517R–expressing ones (Fig 2B). These results suggest that the L517R mutation does not cause impairment of the ion channel activity.

Some pathogenic mutations in *PC2* alter the localization in the cell, including trafficking to primary cilia (Cai et al, 2014; Walker et al, 2019). The overexpression of PC2 variants relative to the endogenous protein levels might result in incorrect ciliary localization. In this study, to characterize the native localization of PC2 mutants in the endogenous cellular context, we obtained *Pkd2*-knockout, *Pkd2* W412G (W414G mutation of human *PC2*), and *Pkd2* L515R (L517R mutation of human *PC2*) mIMCD3 cell lines using the CRISPR/Cas9 technology (Fig S4A–D and Tables S1, S2, and S3). These mutations caused neither changes in the rate of ciliogenesis (56% ± 2%, mean ± SD, WT cells; 55% ± 3%, *Pkd2* L515R cells; 52% ± 4%, *Pkd2* W412G cells) nor perturbations in total cellular cholesterol levels (Fig S4G and H), but these cells with the mutant *Pkd2* gene had intracellular aggregation of cholesterol, suggesting *Pkd2* mutations might impair the intracellular cholesterol trafficking (Fig S5A and B). In *Pkd2*$^{-/-}$ mIMCD3 cells, the PC2 protein was not detected by Western blot analysis (Fig S4E and F), resulting in no detection of ciliary PC2 in immunofluorescence staining (Fig 2C and D). It is known that PC1 and PC2 form a heteromeric polycystin complex (Qian et al, 1997; Tsiokas et al, 1997; Hanaoka et al, 2000), and *Pkd2* knockout inhibited the localization of PC1 to primary cilia in mIMCD3 cells, whereas *Pkd1* knockout disrupted the ciliary localization of PC2 (Fig S5C). Similar to the results in MβCD-treated cells and *Pex14*$^{-/-}$ cells, the PC2 accumulation in primary cilia was reduced in *Pkd2* L515R cells (49% of the level of WT cells) (Fig 2C and D). Although *Pkd2* W414G (mouse *PC2* W412G) had cholesterol-binding ability (Fig 2A), the amount of PC2 W412G in primary cilia was also decreased (55% of the level of WT cells) (Fig 2C and D). Both PC2 mutants showed no accumulation at the ciliary base (Walker et al, 2019) or in the proximal region of cilia (Figs 2C and S5D). The ciliary localization of PC1 was also affected in a similar fashion to PC2 mutants (Fig S5E and F). Given that WT PC2 localizes to primary cilia in a cholesterol-dependent manner (Fig 1), these results imply that the ciliary cholesterol level might somehow be affected by *PC2* mutations.

To clarify how the cholesterol-binding ability of PC2 is involved in modulation of its localization, we simultaneously analyzed the changes in the ciliary cholesterol and PC2 accumulation in *PC2*-mutated cells (Fig 3A–C). We found that there was a linear relationship between the ciliary cholesterol level and amounts of PC2 localization in WT cells (correlation coefficient *r* = 0.66) (Fig 3D and E). When cholesterol in the primary cilia was significantly reduced by MβCD treatment, the PC2 accumulation in primary cilia decreased as if adapting to the lowered cholesterol level (correlation coefficient *r* = 0.66) (Fig 3D and E). As expected, mIMCD3 cells expressing PC2 L515R or PC2 W412G had lower amounts of ciliary cholesterol and PC2 than the untreated WT cells (correlation coefficient *r* = 0.54 in *Pkd2* L515R cells; *r* = 0.66 in *Pkd2* W412G cells) (Fig 3D and E).

As shown in Fig 1, although the transport of cholesterol into the ciliary membrane was inhibited in the *Pex14*$^{-/-}$ mIMCD3 cells, the abundance of PC2 was restored to the untreated WT level in response to the exogenous elevation of ciliary cholesterol, maintaining their linear relationship (correlation coefficient *r* = 0.66 in *Pex14*$^{-/-}$ cells; *r* = 0.62 in cells with cholesterol addition) (Fig 3D and E). The most notable result was that the effects produced by the addition of water-soluble cholesterol strongly depended on the cholesterol-binding ability of PC2. In PC2 W412G, which had the cholesterol-binding ability, the treatment with cholesterol improved the ciliary localization (correlation coefficient *r* = 0.65), in that PC2 W412G was distributed broadly along the entire primary cilia (Figs 3D and E and S5G). In contrast, an increase in ciliary cholesterol was obtained by the addition of cholesterol in *Pkd2* L515R cells, but did not produce noticeable effects (Fig 3A–C). As a result, the correlation between the amounts of ciliary cholesterol and PC2 L515R was completely lost (correlation coefficient *r* = 0.10) (Fig 3D and E). These results indicate that the cholesterol-binding site of polycystin-2 is essential for ciliary localization.

The tubby family protein Tulp3 is required for the ciliary trafficking of polycystins (Hwang et al, 2019). PC2 W414G (mouse PC2 W412G) was able to interact with Tulp3, as well as WT PC2 and PC2 L517R (mouse L515R) (Figs 4A and S6A). To examine whether the Tulp3-mediated ciliary entry of PC2 is involved in the effects of exogenous cholesterol on ciliary localization of the *Pkd2* W414G mutant, we obtained *Tulp3*-knockout and *Pkd2* W412G/*Tulp3*-knockout mIMCD3 cell lines using the CRISPR/Cas9 technology (Fig S6B and C). The depletion of the *Tulp3* gene caused a decrease not only in the ciliary amount of WT PC2, but also in ciliary cholesterol (Fig 4B). The reduction of ciliary localization of WT PC2 in *Tulp3*-knockout cells was not altered even in the presence of sufficient exogenous cholesterol, suggesting that Tulp3 might be epistatically upstream or in the parallel position of cholesterol-dependent ciliary PC2 localization (Fig 4B–D). As expected, the restoration of PC2 W412G localization in primary cilia, induced by externally applying water-soluble cholesterol, was disrupted in *Pkd2* W412G/*Tulp3*-knockout mIMCD3 cells (Figs 4B–D and S6D and E). Given the results showing that trafficking of PC2 mutants into the ciliary membrane (Figs 2C and 3A) depends on the Tulp3-mediated mechanism (Fig 4B), these findings imply that the cholesterol–polycystin-2 interaction is necessary for stable ciliary accumulation rather than ciliary entry.

### Defects in the interaction of polycystin-2 with cholesterol cause polycystic kidney disease

We evaluated whether the disruption of the cholesterol-binding site affects the epithelial lumen architecture. As mentioned above, we first used in vitro spheroidgenesis analysis for *Pkd2*-mutated mIMCD3 cells as a three-dimensional culture model of polycystic kidney disease (Fig S2C). The spheroid formation occurred in both *Pkd2* L515R and W412G cells (Fig 5A), but the appearance of spheroids with a single lumen, in which the primary cilia protruded from the apical surface, was decreased (Fig S7A). Only spheroids with a single lumen were analyzed (Fig S7A and B). The analysis of the relationship between the sizes of lumen and spheroid revealed lumen expansion, involving a marked increase in the ratio of lumen to spheroid area (0.11 ± 0.04, mean ± SD, WT; 0.25 ± 0.05, *Pkd2* L515R; 0.25 ± 0.05, *Pkd2* W412G) (Fig 5B–D). The rate of lumen expansion in these mutant cells was comparable to that of MβCD-treated WT cells and *Pex14*$^{-/-}$ cells. We next investigated the effects of exogenous cholesterol on the epithelial

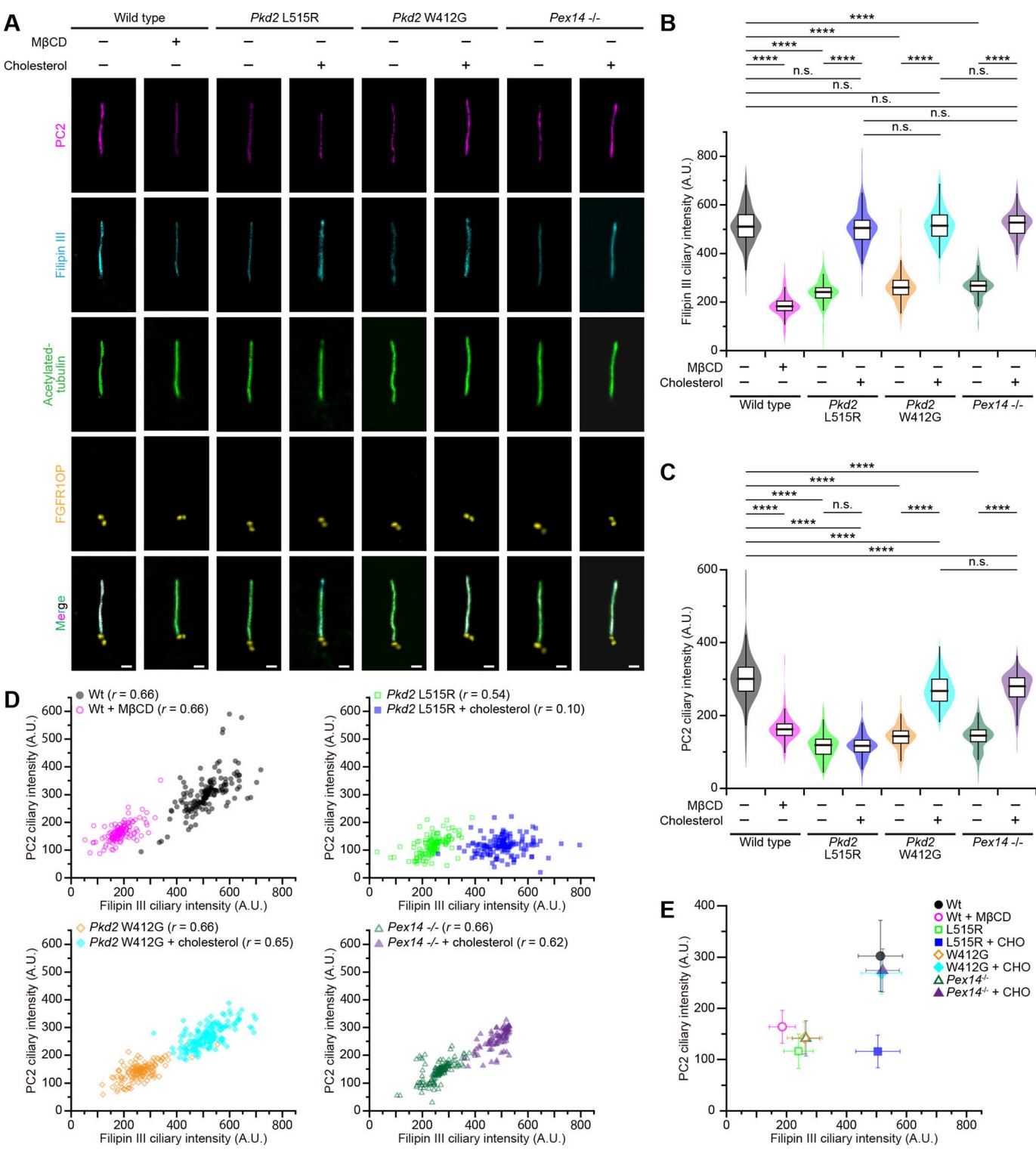

**Figure 3. Localization of polycystin-2 to primary cilia correlates with the amount of ciliary cholesterol.**
**(A)** Ciliary localization of PC2 variants in quiescent $G_0$-phase mIMCD3 cells in a manner dependent on cholesterol. Cells were stained with filipin III (cyan), anti-PC2 (magenta), anti-acetylated tubulin (green), and anti-FGFR1OP (yellow) antibodies. Line-scan measurements of PC2 and filipin III intensity of each primary cilium are shown in Fig S5G. Scale bars, 1 $\mu$m. **(B, C)** Quantification of the filipin III intensity (B) and the PC2 intensity (C) at primary cilia in *Pkd2*-mutated cells. WT cells were treated with or without M$\beta$CD. *Pkd2* L515R cells, *Pkd2* W412G cells, and *Pex14*[−/−] cells were treated with water-soluble cholesterol (****$P < 0.0001$; n.s., not significant, one-way ANOVA with Tukey's multiple comparison tests, n = 3 independent experiments: 49–51 cells per experiment). The graph shows the median and quartiles. **(D, E)** Relationship between the ciliary PC2 and filipin III intensity. **(D)** $r$ indicates the correlation coefficient (D). **(E)** Plots represent mean values (error bars, $\pm$ SD) (E).

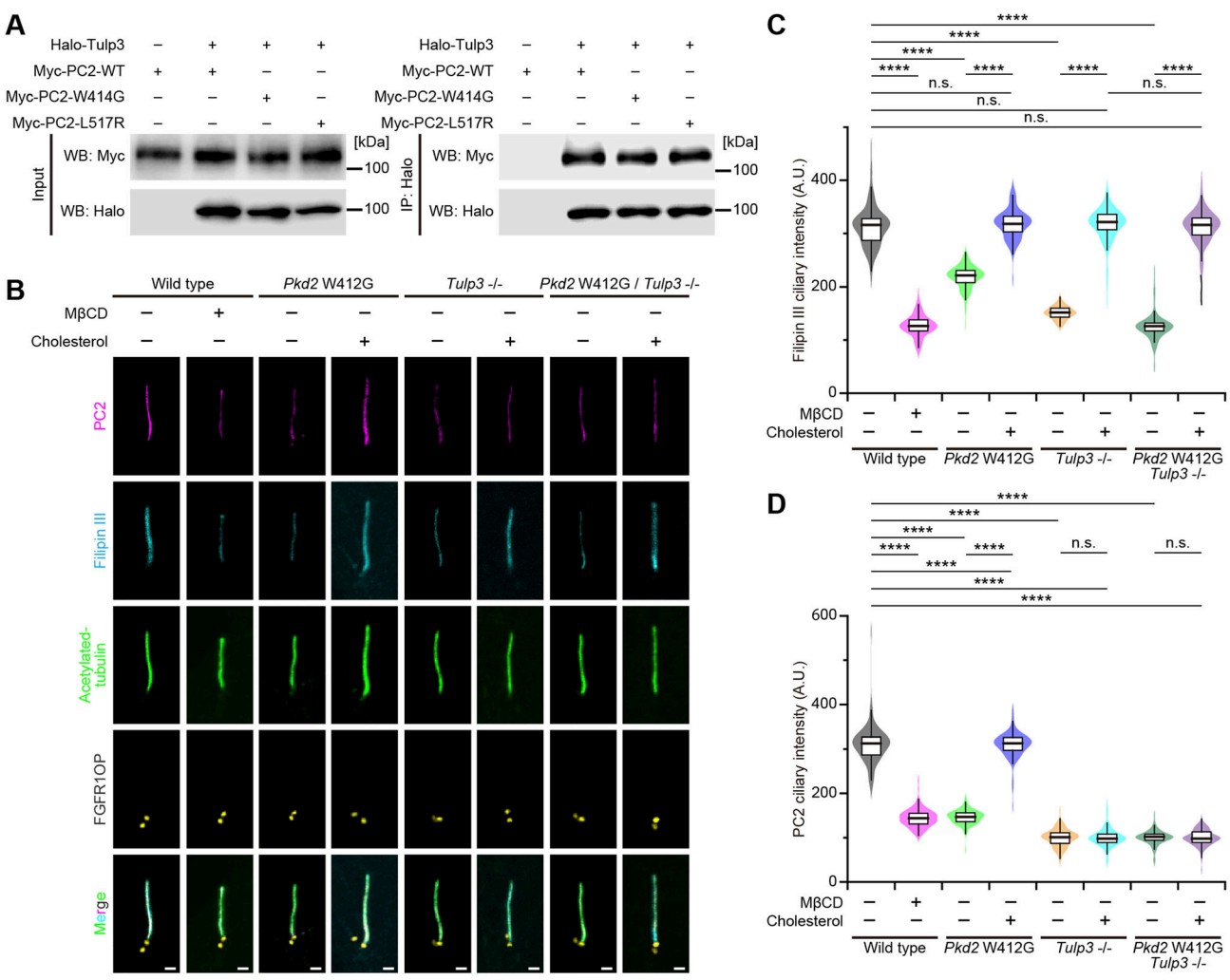

**Figure 4. Ciliary localization of polycystin-2 and the amount of ciliary cholesterol depend on Tulp3-mediated trafficking.**
**(A)** Western blot analysis for the immunoprecipitation assay. The lysates of HEK293T cells transiently expressing either Myc-tagged PC2 W414G mutant or Halo-tagged Tulp3 protein or both proteins were precipitated with anti-Myc antibody–conjugated beads and analyzed by Western blotting using anti-Halo antibody. **(B)** Ciliary localization of PC2 variants in quiescent $G_0$-phase mIMCD3 cells in a manner dependent on cholesterol. Cells were stained with filipin III (cyan), anti-PC2 (magenta), anti-acetylated tubulin (green), and anti-FGFR1OP (white) antibodies. Scale bars, 1 $\mu$m. **(C, D)** Quantification of the filipin III intensity (C) and the PC2 intensity (D) at primary cilia in *Tulp3*-mutated cells. WT cells were treated with or without M$\beta$CD. *Pkd2* W412G cells, *Tulp3*$^{-/-}$ cells, and *Pkd2* W412G/*Tulp3*$^{-/-}$ cells were treated with water-soluble cholesterol (****$P < 0.0001$; n.s., not significant, one-way ANOVA with Tukey's multiple comparison tests, n = 3 independent experiments: 48–52 cells per experiment). The graph shows the median and quartiles.

lumen size by externally applying water-soluble cholesterol to the formed spheroids. The results showed that the ratio of lumen to spheroid area in *Pkd2* W412G cells almost recovered to that of the WT cells without the changes in the cell number (0.11 ± 0.02, mean ± SD) (Fig 5D). However, the cholesterol treatment did not induce the restoration of a lumen-to-spheroid ratio in *Pkd2* L515R cells (0.27 ± 0.05, mean ± SD) (Fig 5D), implying that the low abundance of ciliary PC2 caused by the L515R mutation might contribute to the pathogenesis of this mutation in ADPKD, even in the presence of sufficient ciliary cholesterol.

To apply a reverse genetics approach to confirm that defects in the interaction between cholesterol and PC2 cause polycystic kidney diseases, we attempted to establish *Pkd2* L515R knock-in mice using genome editing technology with ssODN, which can establish heterozygous cell clones using fertilized mouse embryos. A heterozygous (*Pkd2*$^{L515R/+}$) $F_0$ male mouse was generated by

microinjection into the pronucleus of fertilized eggs of C57BL/6J with CRISPR/Cas9 and ssODN (Table S4). The ssODN knock-in *Pkd2*$^{L515R/+}$ male was mated with female C57BL/6J mice to obtain the *Pkd2*$^{L515R/+}$ $F_1$ generation. These offspring were intercrossed to generate homozygous (*Pkd2*$^{L515R/L515R}$) mice, but these showed embryonic lethality (Table S5). We therefore analyzed *Pkd2*$^{L515R/L515R}$ embryo and histological sections in *Pkd2*$^{L515R/L515R}$ kidney, which revealed the enlargement of both Bowman's space and the renal tubule lumen, along with renal cysts (Fig 6A–F). Likewise, in previous reports on *Pkd2* mutant mice (Wu et al, 2000; Walker et al, 2019), *Pkd2*$^{L515R/L515R}$ mice showed heterotaxy, edema, or hemorrhage (Figs 6G–I and S8A and B). Taking these findings together, deficiency or incorrect interaction of peroxisome-derived cholesterol in cilia causes the mislocalization of PC2, resulting in ciliopathy represented by polycystic kidney.

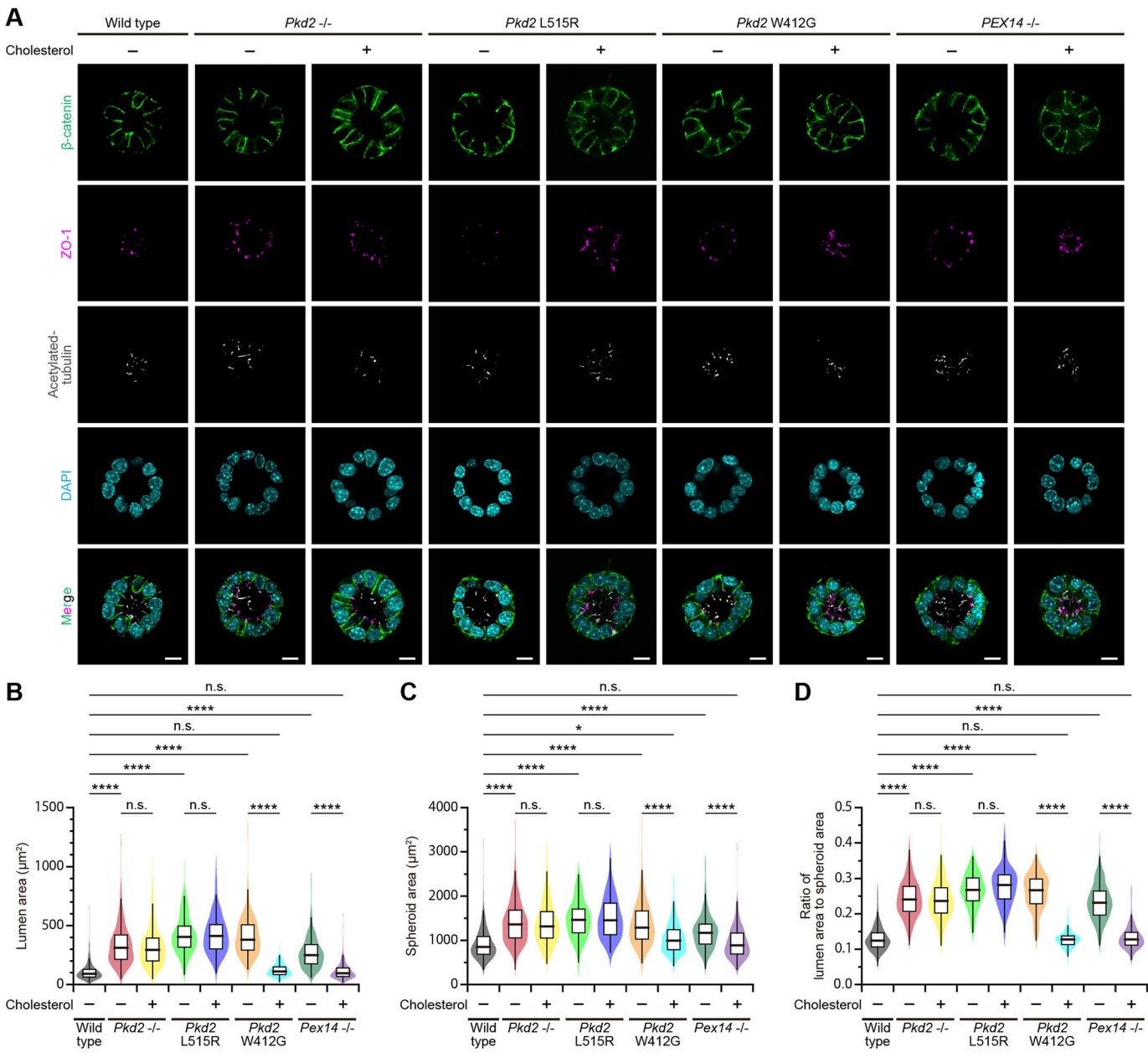

**Figure 5. Cholesterol administration restores the epithelial lumen architecture.**
**(A)** Spheroid formation in *Pkd2*-mutated mIMCD3 cells grown in three-dimensional culture. After the formation of spheroids, cells were treated with or without cholesterol and then stained for β-catenin (green), ZO-1 (magenta), acetylated tubulin (white), and DAPI (cyan). Scale bars, 10 μm. **(B)** Lumen size in *Pkd2*-mutated cells with or without the cholesterol treatment (****$P < 0.0001$; n.s., not significant, one-way ANOVA with Tukey's multiple comparison tests, n = 3 independent experiments: 52–140 spheroids with a single lumen per experiment). The graph shows the median and quartiles. **(C)** Spheroid size in *Pkd2*-mutated cells with or without the cholesterol treatment (****$P < 0.0001$; n.s., not significant, one-way ANOVA with Tukey's multiple comparison tests, n = 3 independent experiments: 52–140 spheroids with a single lumen per experiment). The graph shows the median and quartiles. **(D)** Ratio of lumen area to spheroid area in *Pkd2*-mutated cells with or without the cholesterol treatment (****$P < 0.0001$; n.s., not significant, one-way ANOVA with Tukey's multiple comparison tests, n = 3 independent experiments: 52–140 spheroids with a single lumen per experiment). The graph shows the median and quartiles.

## Discussion

For primary cilia to sense extracellular mechanical and chemical cues, it is thought that both protein and lipid compositions in the ciliary membrane must be segregated from those of the other regions of the plasma membrane (Garcia et al, 2018). In kidney epithelial cells, PC2 localizes to the plasma membrane, the intercellular adhesion structures, and the ER (Koulen et al, 2002; Scheffers et al, 2002; Sammels et al, 2010), whereas cholesterol is distributed in the plasma membrane and intracellular organelle membranes (Ikonen, 2008); both PC2 and cholesterol are strongly enriched in the ciliary membrane (Chailley et al, 1983; Nauli et al,

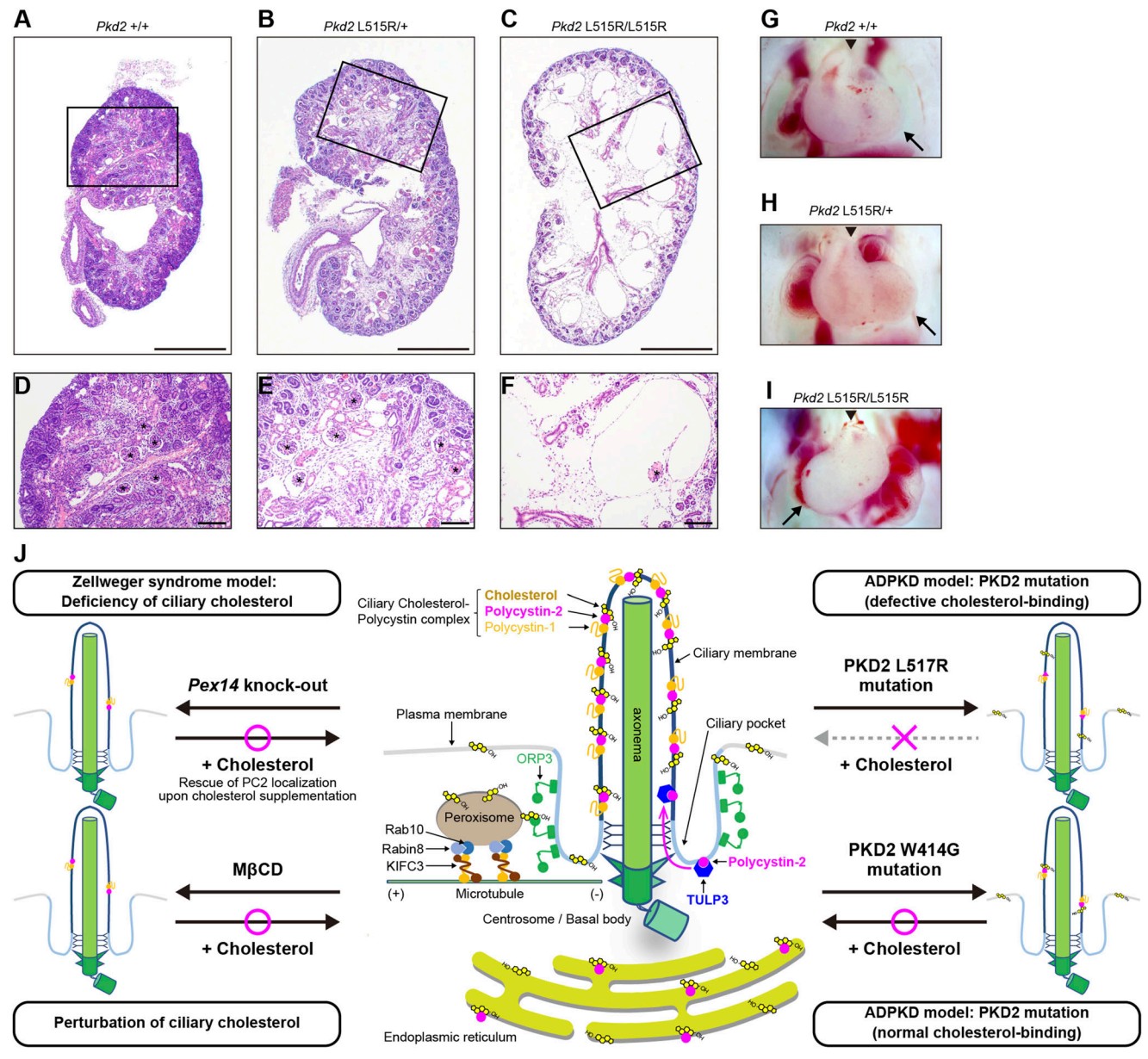

**Figure 6. Defects of cholesterol and PC2 interaction cause polycystic kidney diseases and heterotaxy.**
(A, B, C, D, E, F) HE staining showed mouse fetus kidneys from *Pkd2*$^{+/+}$ (A), *Pkd2*$^{L515R/+}$ (B), and *Pkd2*$^{L515R/L515R}$ (C). Scale bars represent 500 μm. (A, B, C, D, E, F) Boxed areas in (A, B, C) are presented in (D, E, F). (F) Bowman's lumen (asterisk) was notably enlarged in *Pkd2*$^{L515R/L515R}$ (F). Scale bars represent 100 μm. (G, H, I) Stereomicroscopic images show mouse fetus hearts from *Pkd2*$^{+/+}$ (G), *Pkd2*$^{L515R/+}$ (H), and *Pkd2*$^{L515R/L515R}$ (I). (I) Dextrocardia was observed in *Pkd2*$^{L515R/L515R}$ (I). The arrow and arrowhead indicate the apex of the heart and the outflow tract, respectively. (J) Model for the organization of the ciliary cholesterol–polycystin complex. Cartoon summarizing the implications of the current results and previous studies associated with the ciliary cholesterol and polycystin complex. Peroxisomes supply cholesterol into the ciliary membrane via the peroxisomal Rab10/Rabin8/KIFC3 complex and ORP3 at the ciliary pocket. Pex14 knockout or MβCD-mediated cholesterol depletion impairs ciliary localization. Cholesterol rescues the ciliary PC2 in the Pex14 mutant cells, but not the ciliary localization of PKD2 L517R exhibiting deficiency of cholesterol binding. Cholesterol enables PKD2 W414G, which is defective in ciliary localization but exhibits normal cholesterol binding, to exert its function in the ciliary membrane. Overall, the defective interaction between PC2 and cholesterol causes polycystic kidney in humans and mice.

2003; Miyamoto et al, 2020). Notably, patients with Smith–Lemli–Opitz syndrome, which is caused by germline mutations in the *DHCR7* gene encoding the last enzyme in the de novo cholesterol synthesis pathway, show polycystic kidney (Fitzky et al, 1998; Wassif et al, 1998; Nowaczyk & Irons, 2012). A peroxisome biogenesis disorder, ZS, is also characterized by ciliary cholesterol insufficiency and polycystic kidney (Miyamoto et al, 2020). Peroxisomes move along the microtubules to the basal bodies in a peroxisomal Rab10/Rabin8/KIFC3 complex–dependent manner (Miyamoto et al, 2020). At the ciliary pocket, oxysterol-binding protein–related protein 3 (ORP3) mediates cholesterol trafficking from peroxisomes to ciliary membrane (Miyamoto et al, 2020). When zebrafish embryos are treated with statins, which inhibit the rate-limiting enzyme HMG-CoA reductase in the cholesterol synthesis pathway, they develop

ciliopathy spectrum disorders, including situs inversus and poly-cystic kidney (Maerz et al, 2019). However, the mechanistic link between PC2 and cholesterol in polycystic kidney diseases remains unclear. Here, we found that methyl-β-cyclodextrin–mediated lowering of ciliary cholesterol interferes with the ciliary recruitment of PC2 to expand the epithelial lumen and that administering cholesterol to *Pex14*-knockout mIMCD3 cells with inhibited peroxisome-mediated ciliary cholesterol trafficking restores ciliary PC2 localization and lumen size control (Fig 1). Notably, the L517R mutation of *PC2* (L515R mutation of mouse *PC2*) at the cholesterol-binding pocket detected in an ADPKD patient impairs the PC2 properties including the affinity to cholesterol and ciliary locali-zation (Fig 2). Consistent with the in vitro data set, the *Pkd2* L515R mutant mice show polycystic kidney (Fig 6). These findings suggest that cholesterol controls the ciliary function of PC2 to prevent polycystic kidney.

How does cholesterol control the ciliary localization of PC2? It was previously demonstrated that PC2 is mediated by the tubby family protein TULP3, which is a key adaptor of the intraflagellar transport complex A (IFT-A), to be trafficked into the ciliary membrane (Badgandi et al, 2017; Hwang et al, 2019; Legue & Liem, 2019). The tubby domain of TULP3 is anchored to the plasma membrane by phosphatidylinositol 4,5-bisphosphate (PI(4,5)P$_2$) concentrated in the ciliary pocket to capture the PC2 protein around cilia (Badgandi et al, 2017). After reaching cilia, the lower level of PI(4,5)P$_2$ in the ciliary membrane might dislodge PC2 from the tubby domain, thereby concentrating PC2 in the ciliary compartment. Indeed, mice with kidney-specific *Tulp3* knockout showed a lack of ciliary PC2 to polycystic kidneys (Hwang et al, 2019; Legue & Liem, 2019). The variants associated with maintained channel function but dampened ciliary localization, including *PC2* W414G (mouse *PC2* W412G) (Cai et al, 2014) and *PC2*$^{lrm4}$ (mouse *PC2* E442G) (Walker et al, 2019) with mutations in the TOP domain encoded by the S1–S2 loop, cause polycystic kidney in human and mice, respectively. It was also reported that the PC2$^{lrm4}$ mutant protein is unable to pass beyond the distal appendages of the basal body, thereby lowering the level of the PC2$^{lrm4}$ mutant protein in the ciliary membrane (Walker et al, 2019). The PC2$^{lrm4}$ mutant protein might be dampened in the TULP3-mediated ciliary trafficking. In contrast, a minor population of the PC2 W412G mutant protein passes beyond the basal body to enter the ciliary membrane compartment, thereby lowering the level of the PC2 W412G mutant protein in ciliary membranes compared with that of the normal PC2 protein (Fig 2). In this study, we demon-strated that the PC2 W414G mutant protein still interacts with TULP3 and WT of PC2 and that depletion of *Tulp3* in *Pkd2* W412G mutant cells suppresses the ciliary entry of its mutant protein (Fig 4). Further studies are needed to clarify how does the phenotypic difference in the ciliary entry of PC2 mutant proteins appear among the *PC2* mutations within the TOP domain. Interestingly, we found that the ciliary membrane cholesterol level is also significantly reduced in *Pkd2* W412G mutant cells and *Pkd2*-knockout cells, suggesting that the accumulation of PC2 and cholesterol in ciliary membranes is due to an interdependent relationship between them. PC2 is known to localize to not only primary cilia but also ER (Koulen et al, 2002). This study revealed that intracellular choles-terol aggregates outside primary cilia in *Pkd2* W412G mutant cells and *Pkd2*-knockout cells (Fig S5A and B), suggesting that dampened

intracellular cholesterol trafficking up to the cilia in the *Pkd2* defective cells reduces the level of ciliary cholesterol. Notably, this study also reveals that cholesterol supplementation enhances the localization of the PC2 W412G mutant protein to the ciliary mem-brane (Fig 3), implying that cholesterol might promote the entry of PC2 into the ciliary membrane compartment. However, because the major fraction of the PC2 W412G mutant protein is unable to localize into the ciliary membrane even in the presence of a sufficient amount of endogenous cholesterol, despite its ability to bind to cholesterol, other mechanisms for cholesterol-mediated PC2 ciliary localization than enhancing the ciliary entry of PC2, such as TULP3-mediated trafficking, are postulated. Another possibility is that cholesterol controls the conformation of PC2 to enable its accu-mulation in the ciliary membrane. Indeed, in contrast to the PC2 W412G mutant protein, the cholesterol-binding-defective PC2 L515R mutant protein failed to accumulate in the ciliary membrane, even when there was sufficient exogenous cholesterol in the primary cilia (Fig 3). This suggests that cholesterol might capture PC2 in the ciliary membrane compartment to reduce the mobility of the cholesterol–polycystin complex for the accumulation of the PC2 in the ciliary membrane (Fig 6J). In the context of cilium-related Shh signaling, cholesterol directly binds to a seven-pass transmem-brane protein Smoothened (Smo) to change the latter's confor-mation, enabling Smo to accumulate in the ciliary membrane and thereby activating transduction (Huang et al, 2016; Radhakrishnan et al, 2020). Further comparison of the conformations between WT PC2 and PC2 L515R mutant proteins should reveal the mechanism of cholesterol-mediated ciliary localization of PC2. Against this background, ciliary cholesterol may bind directly to target mem-brane proteins for ciliary localization and functional activation.

Whether does cholesterol control the channel activity of PC2? A previous structural and simulation study revealed a cholesterol-binding site located between the S3/S4 helices of the VSLD and channel-pore domain S6 helix of PC2 (Wang et al, 2020). Here, we demonstrate that the *PC2* L517R mutation located at this site does not significantly impair the channel activity despite the lack of cholesterol binding and that *Pkd2* L515R mutant mice show dis-rupted left–right asymmetry determination and polycystic kidney. These findings suggest that cholesterol bound to the site located at PC2 L517 regulates PC2 localization to cilia rather than ion channel activity in vitro and in vivo (Figs 2 and 6). It is possible that ion channel activity detected in the PC2 L517R mutant protein was due to the presence of other sterol-binding sites required for ion channel activity. Indeed, it was reported that an oxysterol enriched in cilia, namely, 7β,27-dehydrocholesterol (7β,27-DHC), binds to the distinct pocket site formed by the pre-S1 helix and S4 and S5 linkers of PC2 to control the ion channel activity in vitro (Ha et al, 2024). Given the structural similarity between cholesterol and 7β,27-DHC, both molecules could bind to these pockets of PC2 to promote the ciliary PC2 accumulation and ion channel activity, suggesting that sterols can enhance the ciliary polycystin channel activity. In addi-tion, it was reported that ciliary localization of the 7β,27-DHC-binding-site PC2 mutant protein is not affected (Ha et al, 2024). However, because the data were in cells overexpressing those mutants, they may not reflect localization under physiological conditions. That is, it remains possible that the 7β,27-DHC-binding site is also required for PC2 ciliary localization. Further studies are necessary to clarify the

conformational basis for the increase in the ion current and ciliary localization of PC2 by cholesterol to develop unique agonists for the polycystin complex. In the ADPKD model mice, genetic manipulation of *PKD2* gene re-expression restores polycystic kidneys, implying that the epithelial phenotypes of ADPKD, including the expanded lumen size and isolated cysts, are reversible and that polycystin complex activators are potential unique therapeutics for ADPKD (Dong et al, 2021). Our results suggest that ciliary cholesterol is a potential therapeutic target to compensate for the loss-of-function mutations of the *PKD2* gene underlying ADPKD.

In conclusion, we demonstrate that peroxisome-mediated cholesterol trafficking is essential for the ciliary localization of the polycystin complex to prevent the occurrence of polycystic kidney.

# Materials and Methods

### Cell cultures

mIMCD3 (CRL-2123; ATCC), HEK293 (CRL-1573), and HEK293T (CRL-3216) cells were cultured in DMEM (Gibco, Thermo Fisher Scientific) supplemented with 10% FBS and 25 $\mu$g/ml gentamycin. All cell lines were maintained in humidified air with 5% $CO_2$.

### Gene targeting in mIMCD3 cells using CRISPR/ObLiGaRe

A total of $2 \times 10^5$ mIMCD3 cells were seeded into one well of a six-well plate 24 h before lipofection. A total of 20 ng of targeting vector and 600 ng of the *pX330* plasmid containing a guide RNA sequence for targeting were cotransfected into cells using Lipofectamine LTX (Thermo Fisher Scientific) following the manufacturer's protocol. Forty-eight hours later, cells were reseeded into 15-cm dishes and cultured in 10% FBS/DMEM containing 2 mg/ml G418 (Nacalai Tesque). Drug-resistant cell colonies were picked up on days 14–18 after transfection. These colonies were divided into two aliquots for genotyping and clonal expansion. PCR genotyping was performed using three types of primer pairs: the first primer pair for detecting the target gene locus (Table S1), the second primer pair consisting of the forward primer in the target gene locus and the *Neo*[r]-reverse primer (5'-GCGGATCT-GACGGTTCACTAAACCAGC-3') for detecting the forward insertion of the drug-resistant gene cassette, and the third primer pair consisting of the reverse primer in the target gene locus and the *Neo*[r]-reverse primer for detecting the reversed insertion, in accordance with a previous study (Miyamoto et al, 2020). The presence or absence of insertion or deletion was determined by direct sequencing.

### Generation of *Pkd2*-edited mIMCD3 cells

A total of 200 pmol of 150-mer ssODN carrying *Pkd*2 variants and a silent Afe I site (Fig S4B) and 1 $\mu$g of the *pX459* containing guide RNA sequences were cotransfected into $1 \times 10^6$ mIMCD3 cells with kit R (Lonza) and program U-017 (Nucleofector 2b device; Lonza) following the manufacturer's protocol.

### Immunofluorescence microscopy

To detect the protein epitope, cells grown on coverslips (Matsunami) were fixed in 4% PFA at RT for 15 min. After fixation, cells were permeabilized in 0.2% Triton X-100 at RT for 10 min or 100% methanol at −20°C for 5 min. Briefly, cells were washed with PBS three times and blocked with 1% BSA/PBS at RT for 30 min. After the blocking step, the samples were incubated with primary antibodies (mouse anti-PC1 monoclonal IgG$_1$Ab [sc130554]; Santa Cruz Biotechnology; mouse anti-PC2 monoclonal IgG$_{2b}$Ab [sc28331]; Santa Cruz; mouse anti-acetylated tubulin monoclonal IgG$_{2b}$Ab [T7451]; Sigma-Aldrich; rabbit anti-ARL13b pAb [17711-1-AP]; Proteintech; mouse anti-γ-tubulin monoclonal IgG$_1$Ab [T6557]; Sigma-Aldrich; rabbit anti-FGFR1OP pAb [11343-1-AP]; Proteintech) diluted with 1% BSA/PBS or Can Get Signal immunostain solution A (Toyobo) at RT for 60 min. To detect intracellular cholesterol, serum-starved cells were fixed in 4% PFA at RT for 15 min and blocked with 1% BSA/PBS at RT for 30 min. Blocked cells were incubated with 50 $\mu$g/ml filipin III (Cayman Chemical) in Can Get Signal immunostain solution A and each antibody for counterstaining at RT for 60 min. After three washes with PBS, cells were incubated with Alexa Fluor 405–, Alexa Fluor 488–, Alexa Fluor 546–, Alexa Fluor 594–, or Alexa Fluor 647–conjugated goat or guinea pig secondary antibodies against rabbit IgG, rat IgG, or mouse IgG subclasses (Thermo Fisher Scientific) at RT for 30 min. The cells were washed three times with PBS and mounted with ProLong Diamond antifade medium (Thermo Fisher Scientific). The stained cells were examined under an LSM800 confocal microscope (Carl Zeiss) or FV3000 confocal microscope (Olympus). The measurement conditions (laser power, gain for each detector, etc.) were kept consistent across experiments to compare fluorescence intensity. ImageJ (https://imagej.net/ij/index.html) was used to analyze microscope images and measure the fluorescence intensity after subtracting the background.

### Cholesterol depletion and supplementation experiments

Cells at 45% confluence were incubated in serum-free DMEM for 48 h to induce ciliogenesis. To remove intracellular cholesterol, serum-starved mIMCD3 cells were treated with 1.5% methyl-$\beta$-cyclodextrin (Sigma-Aldrich) in DMEM for 1.5 h. For the cholesterol complementation assay, the cells were incubated with 50 $\mu$M water-soluble methyl-$\beta$-cyclodextrin–cholesterol complex (Sigma-Aldrich) in serum-free DMEM for 2 h. After washing with serum-free DMEM, the cells were cultured for 6 h.

### Live-cell imaging

WT mPkd2, the C terminus of which was tagged with *A. coerulescens* green fluorescent protein (AcGFP), and DsRed2-PACT (Miyamoto et al, 2020) were cotransfected into $1 \times 10^5$ *Pkd2*-knockout mIMCD3 cells with Lipofectamine LTX (Thermo Fisher Scientific) in a glass-base dish (AGC Techno Glass). At 24 h after transfection, the medium was replaced with serum-free DMEM, and then, cells were incubated for 24 h to induce ciliogenesis. A glass-base dish was placed in a moisture chamber of the microscope stage (Tokai Hit) maintained at 37°C in humidified air with 5% $CO_2$. Live-cell imaging of PC2-AcGFP and DsRed2-PACT was performed with an LSM800 confocal microscope (Carl Zeiss). The

z-stack images were collected at 2- to 4-min intervals during observation.

## 3D spheroid model assay

A total of 100 $\mu$l of 1 × 10$^5$ cells/ml WT mIMCD3 or mutant lines was mixed with 100 $\mu$l of Matrigel (Corning) and seeded into one well of the eight-well-chambered coverglass. After 30 min of incubation at 37°C, 200 $\mu$l of 10% FBS/DMEM was applied into each well and cells were cultured for 72 h. After 72 h, the culture medium was aspirated and serum-free DMEM was added to induce ciliogenesis. After 24 h of culture in serum-free DMEM, the cells were treated with 50 $\mu$M water-soluble methyl-$\beta$-cyclodextrin–cholesterol complex (Sigma-Aldrich) in DMEM for 3 h. After washing three times, cells were cultured for 8 h. To observe spheroidgenesis, the cells were washed with PBS and fixed in 4% PFA at RT for 30 min. After five washes with PBS, the cells were blocked in PBS containing 7 mg/ml gelatin and 0.5% Triton X-100 (blocking buffer) at RT for 30 min. Then, the cells were treated with primary antibodies (mouse anti-acetylated tubulin mAb [T7451]; Sigma-Aldrich; rat anti-ZO-1 mAb [sc-33725]; Santa Cruz; rabbit anti-$\beta$-catenin pAb [71-2700]; Thermo Fisher Scientific) diluted with blocking buffer at 4°C for overnight. After primary antibody treatment, the cells were washed with blocking buffer five times and incubated with Alexa Fluor 488–, Alexa Fluor 594–, or Alexa Fluor 647–conjugated goat or guinea pig secondary antibodies (Thermo Fisher Scientific) and DAPI (Dojindo) at RT for 4 h. Then, the cells were washed with blocking buffer three times and PBS twice and mounted with VECTASHIELD Mounting Medium (Vector Laboratories). The spheroids were examined by obtaining a confocal z-stack with an LSM800 confocal microscope (Carl Zeiss) or FV3000 confocal microscope (Olympus) (Fig S2F). The number of lumens inside a spheroid was identified by analyzing z-stack images of ZO-1 and $\beta$-catenin. Then, only in spheroids with a single lumen, the largest cross-sectional area (the xy-plane) in the z-stack of each spheroid was determined by analyzing images of $\beta$-catenin. The spheroid area ($\beta$-catenin) and lumen area (ZO-1) were quantified in the same xy-plane with the largest cross-sectional area of the spheroid.

## Immunoprecipitation and Western blot analyses

To examine the interaction between Halo-tagged Tulp3 and Myc-tagged hPKD2 (WT, W414G, and L517R mutant), HEK293T cells were cotransfected with expression vectors using Lipofectamine LTX (Thermo Fisher Scientific) in a six-well plate and cultured for 48 h. After two washes with PBS, the cells were lysed in lysis buffer (20 mM Tris–HCl, pH 7.4, 200 mM NaCl, 2.5 mM MgCl$_2$, 0.5% Nonidet P-40) containing protease inhibitor cocktail (Nacalai Tesque). The lysates were homogenized with a 21-gauge needle and incubated on ice for 15 min. After centrifugation at 20,400$g$ for 15 min at 4°C, supernatants were collected. For immunoprecipitation analysis, the supernatants were incubated with mouse anti-Halo-tag mAb and protein A/G agarose beads (Santa Cruz) for 16 h at 4°C, and the beads then were washed five times with washing buffer (20 mM Tris–HCl, pH 7.4, 200 mM NaCl, 2.5 mM MgCl$_2$, 0.5% Nonidet P-40). The immunoprecipitates were analyzed by 8% SDS–PAGE and transferred to PVDF membranes for Western blot analyses using each primary

antibody (mouse anti-Myc-tag mAb [M192-3]; MBL; mouse anti-Halo tag mAb [G9211]; Promega; rabbit anti-PEX14 pAb [10594-1-AP]; Proteintech; mouse anti-$\alpha$-tubulin mAb [ab7291]; Abcam; rabbit anti-PC2 pAb [19126-1-AP]; Proteintech; rabbit anti-GFP [598]; MBL; rabbit anti-Tulp3 [13637-1-AP]; Proteintech) and secondary antibodies (peroxidase-labeled anti-mouse antibody [NA931VS]; GE Healthcare, Waukesha, WI, USA; peroxidase-labeled anti-rabbit antibody [NA934VS]; GE Healthcare). Western blot analysis was performed following the procedure in a previous study (Miyamoto et al, 2020).

## Electrophysiology

HEK293 cells stably expressing hPKD2 (WT, W414G, and L517R), the C terminus of which was tagged with *A. coerulescens* green fluorescent protein (AcGFP), were generated. After seeding cells in polymer-coverslip bottom dishes ($\mu$-Dish 35 mm low; ibidi, Nippon Genetics), the dishes were placed on a microscope stage 24 h later for whole-cell recording.

Whole-cell recording was performed on GFP-positive HEK293 cells at RT using an upright microscope (BX51WI; Olympus) equipped with an IR-CCD camera system (R-1000; DAGE-MTI), as reported previously (Matsuoka et al, 2021). The pipette solution was composed of (in mM) 120 KCl, 10 NaCl, 2 MgCl$_2$, 0.5 CaCl$_2$, 10 Hepes, 5 EGTA, and 2 Mg-ATP (pH 7.3, adjusted with KOH) (Pelucchi et al, 2006). The bath was perfused with the normal external solution, which was composed of (in mM) 125 NaCl, 2.5 KCl, 2 CaCl$_2$, 1 MgSO$_4$, 1.25 NaH$_2$PO$_4$, 26 NaHCO$_3$, and 20 glucose, and bubbled with 95% O$_2$ and 5% CO$_2$. The pipette resistance was ~2–4 M$\Omega$. The ramp pulse protocol was applied over a duration of 1 s, ranging from –100 mV to +100 mV from an initial holding voltage of –60 mV. The membrane current was recorded under the voltage clamp mode using DOUBLE IPA (Sutter Instrument Company). The signals were filtered at 3 kHz and digitized at 20 kHz. Online data acquisition and offline data analysis were performed using SutterPatch version 2.3.1 (Sutter Instrument Company) and Igor Pro 8.0.4 (WaveMetrics). Liquid junction potential was not corrected. If the electrode potential polarization at the end of the recording exceeded ±5 mV from the initial value, the recording was omitted from the analyses. Recordings with a series resistance higher than 10 M$\Omega$ were also omitted. Only HEK293 cells with a holding current within ±60 pA at a holding potential of –60 mV were analyzed.

## Cholesterol pull-down assay

A total of 2.5 mg of FG COOH beads (Tamagawa Seiki) was transferred to 1.5-ml tubes and washed with N,N-dimethylformamide (DMF). After three washes, supernatants were removed by centrifugation at 20,400$g$ and RT for 5 min. The cholesterol-conjugated beads were prepared following the manufacturer's protocol. A total of 147 $\mu$l of DMF, 23 $\mu$l of 1 M N,N-diisopropylethylamine, 90 $\mu$l of 100 mM fluoro-N,N,N',N'-tetramethylformamidinium hexafluorophosphate, 150 $\mu$l of 100 mM cholesterol, and 90 $\mu$l of Oxyma Pure were applied to 2.5 mg of the beads. To conjugate cholesterol to the beads, mixtures were incubated at RT overnight. After three washes with DMF, the beads were dissolved with DMF and 100 $\mu$l of 1 M N-hydroxysuccinimide

was added to them. Then, 19.2 mg of 1-(3-dimethylaminopropyl)-3-ethylcarbodiimide hydrochloride was applied to each tube, and mixtures were incubated at RT for 2 h. After three washes with DMF, 500 $\mu$l of 2-aminoethanol was added to the beads, and each tube was incubated at RT for 2 h. After three washes with distilled water, beads were stored at 4°C. To prepare Myc-tagged hPKD2 (WT and L517R mutant), expression vectors were transfected to HEK293T cells using Lipofectamine LTX (Thermo Fisher Scientific) in a six-well plate. Twenty-four hours later, the cells were washed with PBS, lysed in 200 $\mu$l of lysis buffer (20 mM Tris–HCl, pH 7.4, 200 mM NaCl, 2.5 mM $MgCl_2$, 0.5% NP-40), homogenized with a 21-gauge needle, and incubated on ice for 30 min. After centrifugation at $20,400g$ and 4°C for 15 min, supernatants were collected. The collected supernatants were diluted fivefold with dilution buffer (20 mM Tris–HCl, pH 7.4, 200 mM NaCl, 2.5 mM $MgCl_2$). The prepared cholesterol-conjugated beads were washed with washing buffer (20 mM Tris–HCl, pH 7.4, 200 mM NaCl, 2.5 mM $MgCl_2$, 0.1% NP-40) and 200 $\mu$l of diluted lysates was added to the beads. After incubation at 4°C for 4 h, beads were washed with washing buffer. The interaction of cholesterol with Myc-tagged hPKD2 proteins was detected by SDS–PAGE and Western blot analysis using rabbit anti-Myc pAb (M192-3; MBL).

### Mice

Sperm and eggs were collected from C57BL/6J Jcl mice and subjected to in vitro fertilization. The nucleic acid was delivered by electroporation to a fertilized egg of C57BL/6J Jcl or by microinjection into its pronucleus. After the introduction of nucleic acid, the zygotes were incubated overnight. The selected two-cell-stage zygotes were transplanted into pseudopregnant ICR mice. One $F_0$ male pup was identified as WT/KI (*Pkd2* L515R knock-in), and two $F_0$ pups were identified as indel mutants (WT/10 bp del female, WT/1 bp ins male) (Table S4). The heterozygous ($Pkd2^{L515R/+}$) $F_0$ male mouse was then intercrossed as a founder mouse with $Pkd2^{+/+}$ female mice, and the obtained heterozygous mice were mated to create homozygous $Pkd2^{L515R/L515R}$. The kidneys were analyzed after E15.5 with reference to a previous study (Walker et al, 2019).

All animal experiments were approved by the Animal Ethics Committee of Yamaguchi University Graduate School of Medicine (approval number: J21027).

### Statistical analysis

All experiments were performed independently at least three times. The data are shown as the mean ± s.d. unless otherwise stated. Statistical significance was determined using one-way analysis of variance (ANOVA) followed by Tukey's multiple comparison test using Origin Pro software (OriginLab). Differences in mean values were considered statistically significant at $P < 0.05$.

## Data Availability

The data underlying this study are available from the corresponding author upon reasonable request.

## Supplementary Information

## Acknowledgements

We thank Drs. Yohei Katoh, Akihito Inoko, Atsuko H Iwane, Makoto Furutani-Seiki, and Akira Shimamoto for helpful discussions. This work was supported by Grants-in-Aid for Scientific Research from the Ministry of Education, Culture, Sports, Science and Technology of Japan (to T Miyamoto: 20K21845 and 21H02718); AMED-PRIME from the Japan Agency for Medical Research and Development, AMED (to T Miyamoto: JP18gm5910011h0004, 19nk0101509h0002, and 23nk0101562h9903); Chugai Foundation for Innovative Drug Discovery Science (to T Miyamoto); and Astellas Foundation for Research on Metabolic Disorders (to T Miyamoto).

### Author Contributions

T Itabashi: conceptualization, data curation, formal analysis, validation, investigation, visualization, methodology, and writing—original draft, review, and editing.
K Hosoba: data curation, formal analysis, validation, investigation, methodology, and writing—original draft, review, and editing.
T Morita: data curation, formal analysis, validation, investigation, methodology, and writing—original draft, review, and editing.
S Kimura: data curation, formal analysis, validation, investigation, and methodology.
K Yamaoka: data curation, formal analysis, validation, investigation, methodology, and writing—original draft, review, and editing.
M Hirosawa: data curation, formal analysis, investigation, and methodology.
D Kobayashi: data curation, formal analysis, investigation, and methodology.
H Kishi: investigation.
K Kume: investigation and methodology.
H Itoh: supervision and methodology.
H Kawakami: supervision and methodology.
K Hashimoto: supervision, investigation, and writing—original draft, review, and editing.
T Yamamoto: supervision and validation.
T Miyamoto: conceptualization, supervision, funding acquisition, validation, investigation, project administration, and writing—original draft, review, and editing.

### Conflict of Interest Statement

The authors declare that they have no conflict of interest.

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
