## [Reviewer comments · Life Science Alliance]

Life Science Alliance

Cholesterol ensures ciliary polycystin-2 localization to prevent polycystic kidney disease

Takeshi Itabashi, Kosuke Hosoba, Tomoka Morita, Sotai Kimura, Kenji Yamaoka, Moe Hirose, Daigo Kobayashi, Hiroko Kishi, Kodai Kume, Hiroshi Itoh, Hideshi Kawakami, Kouichi Hashimoto, Takashi Yamamoto, and Tatsuo Miyamoto

DOI: <https://doi.org/10.26508/lsa.202403063>

Corresponding author(s): Tatsuo Miyamoto, Yamaguchi University

Review Timeline:

Submission Date:	2024-09-26
Editorial Decision:	2024-11-15
Revision Received:	2025-01-07
Editorial Decision:	2025-01-09
Revision Received:	2025-01-09
Accepted:	2025-01-10

Transaction Report:

November 15, 2024

Re: Life Science Alliance manuscript #LSA-2024-03063-T

Prof. Tatsuo Miyamoto
Yamaguchi University
Department of Molecular and Cellular Physiology, Graduate School of Medicine
Minami-kogushi 1-1-1
Ube, Yamaguchi 755-8505
Japan

Dear Dr. Miyamoto,

Thank you for submitting your manuscript entitled "Cholesterol ensures ciliary polycystin-2 localization to prevent polycystic kidney disease" to Life Science Alliance. The manuscript was assessed by expert reviewers, whose comments are appended to this letter. We invite you to submit a revised manuscript addressing the Reviewer comments.

Thank you for this interesting contribution to Life Science Alliance. We are looking forward to receiving your revised manuscript.

Sincerely,

B. MANUSCRIPT ORGANIZATION AND FORMATTING:

Reviewer #1 (Comments to the Authors (Required)):

Patients with Zellweger syndrome (ZS) often present with ciliopathy-like symptoms, such as polycystic kidney disease and retinitis pigmentosa. The link between peroxisomal dysfunction and cilium-related polycystic kidney disease remains unclear, although previous work by the authors demonstrated that peroxisomes play a role in trafficking cholesterol to the cilia. In this study, they revealed that reducing ciliary cholesterol-either through pharmacological methods or genetic disruption of peroxisomes-impairs the localization of the ciliary protein polycystin-2 (PC2). Using cultured renal cells, they showed that a PC2 variant with a missense mutation in its cholesterol-binding site fails to localize to the ciliary membrane, even when cholesterol is supplied. Furthermore, homozygous mice carrying the same mutation developed ciliopathy-related conditions, including situs inversus and polycystic kidney disease. The manuscript is well written but with issues in figure presentation and concerns in some of the experiments. Overall, the findings are novel, and they shed light on the role of peroxisomal cholesterol transport in the functions of ciliary polycystins, and suggests new avenues for understanding the pathogenesis and potential treatment strategies for ADPKD.

Major:

1. Some of the figure images appear misleading. For instance, in Fig. 1A-C, the text states: "ciliary accumulation of both PC1 and PC2 was inhibited (62% of the level of untreated wild-type cells for PC1, and 50% for PC2)." However, the representative images in Fig. 1A suggest a much more dramatic reduction in PC1 and PC2 within the cilia. A similar discrepancy is observed in other figures, such as Fig. 1D-E. According to Fig. 1E, the average levels of the Filipin III signal in M β CD-treated and Pex14 $^{-/-}$ cells are about 50% of those in untreated or WT cells. Yet, in Fig. 1D, the Filipin III signals appear almost completely absent in these cells.

Additionally, in Figs. 3A and 4A, fluorescence signals (including background noise) seem entirely lost, contrasting sharply with the significant residual signals depicted in Figs. 3B-C and 4B-C. This inconsistency between quantitative data and visual representations raises concerns about the accuracy of the representative images and their alignment with the reported measurements.

2. What is the mechanism underlying the reduction of PC1 and PC2 levels in cilia following a 1.5-hour treatment with methyl- β -cyclodextrin (M β CD)? While the treatment may block the trafficking of new PC1 and PC2 into cilia, it remains unclear what happens to the polycystins already present in the cilia before treatment. Are these pre-existing proteins internalized through endocytosis? If so, it could be valuable to perform live-cell imaging (using fluorescently tagged polycystins) to observe the internalization process during M β CD treatment. This approach would provide direct evidence of whether endocytic pathways are involved in the ciliary clearance of these proteins and offer insights into their fate under cholesterol-depleting conditions.

3. The current study uses the ratio of lumen area to spheroid area in 3D-cultured mIMCD3 cells as a readout for the effects of cholesterol depletion. However, has any prior research demonstrated that lumen size in this type of experiment effectively models kidney cyst formation? Additionally, what is the relationship between lumen size and the function of ciliary polycystins in these spheroids.

These questions raise important points. If lumen size is intended to serve as a proxy for in vivo cyst formation, it would be helpful to reference studies that have validated this model. If no direct links between lumen size and kidney cyst formation have been established, the study should clarify the rationale for using this readout and discuss its relevance to in vivo cyst development.

4. The current study assessed PC2 channel currents by expressing PC2 and its mutants in HEK293 cells. However, previous research has shown that in these cells, PC2 predominantly localizes to the ER membrane rather than the plasma membrane, making its channel activity difficult to record. Given the small recorded currents (Fig. 2B), the authors need to provide stronger evidence that these currents indeed originate from the PC2 channel.

To address this, the following points should be clarified:

- Localization of PC2: Do WT and mutant PC2 traffic to the plasma membrane in HEK293 cells? Immunofluorescence or surface biotinylation assays could help confirm this.
- Baseline controls: What do the currents look like in untransfected or empty vector-transfected cells? These controls are essential to rule out background or nonspecific currents.

- Pathogenic mutations: Testing other well-characterized pathogenic mutations like D511V, which abolishes PC2 channel function, would provide further validation.
- W414G mutation: The effects of W414G on PC2 channel function remain controversial, as shown in studies by Cai et al. (Altered trafficking and stability of polycystins underlie polycystic kidney disease, JCI, 2014) and Wang et al. (The diverse effects of pathogenic point mutations on ion channel activity of a gain-of-function polycystin-2, JBC, 2023). Discussing these differences would enhance the rigor of the findings.

5. What happens to ciliary PC1 in the PC2-L515R and PC2-W412G mutant cells?

Minor:

1. Line 83, the three citations only show the interaction between PC1 and PC2, for key studies showing the 1:3 stoichiometry of the PC1/PC2 complex, two other references should be cited:

- 1) Yu et al., Structural and molecular basis of the assembly of the TRPP2/PKD1 complex, PNAS, 2009
- 2) Su et al., Structure of the human PKD1-PKD2 complex, Science, 2018

2. The authors observed that the reduction in ciliary localization of wild-type PC2 in 233 Tulp3-knockout cells remained unchanged even with sufficient exogenous cholesterol. From this, they concluded that "Tulp3 might be epistatically upstream of cholesterol-dependent ciliary PC2 localization." However, an alternative interpretation could be that both Tulp3 and cholesterol are simultaneously necessary for PC2's trafficking to the cilium. This possibility suggests that these two factors might work in tandem rather than sequentially, with each playing a critical, non-redundant role in facilitating PC2's proper localization.

3. Lines 257-259: "implying that the low abundance of ciliary PC2 due to the L515R mutation causes the pathogenesis of ADPKD, even in the presence of sufficient ciliary cholesterol." The authors might consider softening this interpretation. A revised version could read: "implying that the low abundance of ciliary PC2 caused by the L515R mutation might contribute to the pathogenesis of this mutation in ADPKD, even in the presence of sufficient ciliary cholesterol." This adjustment leaves room for other potential contributing factors and acknowledges the complexity of the disease mechanism.

Reviewer #2 (Comments to the Authors (Required)):

Comments to the Authors:

Please provide your review in the following format:

A short summary of the paper, including description of the advance offered to the field.

The current study examines cholesterol's role in polycystin-2 (PC2) localization to primary cilia and its implications for polycystic kidney disease (PKD). The authors demonstrate that peroxisome-derived cholesterol is essential for proper PC2 localization in the ciliary membrane, and disruption of this interaction leads to an increased cystic index. They identify a critical cholesterol-binding site on PC2 (L517) and claim that mutation of this site (L517R) impairs ciliary localization without affecting channel function. Using both cell culture models and mouse genetics, they show that proper cholesterol-PC2 interaction is necessary to prevent cyst formation. This work provides mechanistic and therapeutic insights into PKD pathogenesis.

For each main point of the paper, please indicate if the data are strongly supportive. If not, explicitly state the additional experiments essential to support the claims made and the timeframe that these would require.

Evaluation of Main Points:

A. Role of cholesterol in PC2 ciliary localization

The authors establish a clear link between cholesterol levels and PC2's presence in primary cilia. The rescue experiments with cholesterol supplementation strongly support their conclusions by restoring PC2 localization when cholesterol levels are replenished.

B. Characterization of PC2 L517R Mutation

Strengths: Biochemical pull-downs and imaging studies provide multiple lines of evidence that support their findings about this mutation's effects.

Concerns: The electrophysiology data in Figure 2B requires several important controls for proper interpretation. Incorporating non-transfected control recordings would help establish baseline currents. Recent work from Ha et al. (2024) indicates that PC2 does not form active channels in HEK cells, and it would be helpful to address how the author's findings relate to their observations.

Given the small current amplitudes (pA/pF range), additional validation would strengthen the conclusions. Did the authors consider using standard TRP channel blockers (gadolinium, ruthenium red) to confirm these are PC2-mediated currents? Additionally, characterization of basic channel properties including reversal potentials and ion selectivity would provide valuable validation of these currents. We need experiments to help distinguish PC2-specific channel activity from background conductances.

C. In Vivo Validation: The generation and characterization of the PC2 L517R knock-in mouse model significantly validate their cellular findings. The mouse phenotypes, including enlarged Bowman's space, renal cysts, and heterotaxy, effectively demonstrate the physiological relevance of cholesterol-PC2 interaction. The correlation between cellular and organismal phenotypes is convincing.

Lastly, indicate any additional issues you feel should be addressed (text changes, data presentation, statistics etc.).

Note: all your comments will be transmitted to the authors and the other referees. Should there be any issues with the manuscript, in particular concerns about ethical standards, data integrity, biosecurity, or conflicts of an academic or commercial nature that need to be communicated to the editor, please contact us directly at contact@life-science-alliance.org

Overall Assessment: Despite the concerns about electrophysiology controls, this study makes a valuable contribution to the field. It establishes a novel mechanistic link between cholesterol metabolism and PKD, potentially opening new therapeutic avenues. With the suggested revisions, particularly to the electrophysiology data, this work merits publication in a good-impact journal.

Responses to Reviewers

We thank the reviewers for their constructive and thoughtful comments. According to the reviewer's comment, we revised the manuscript carefully.

We hope the revision is appropriate and meets your requirements for publication of our manuscript in *Life Science Alliance*.

Our specific responses to which are as follows:

Reviewer #1:

Patients with Zellweger syndrome (ZS) often present with ciliopathy-like symptoms, such as polycystic kidney disease and retinitis pigmentosa. The link between peroxisomal dysfunction and cilium-related polycystic kidney disease remains unclear, although previous work by the authors demonstrated that peroxisomes play a role in trafficking cholesterol to the cilia. In this study, they revealed that reducing ciliary cholesterol-either through pharmacological methods or genetic disruption of peroxisomes-impairs the localization of the ciliary protein polycystin-2 (PC2). Using cultured renal cells, they showed that a PC2 variant with a missense mutation in its cholesterol-binding site fails to localize to the ciliary membrane, even when cholesterol is supplied. Furthermore, homozygous mice carrying the same mutation developed ciliopathy-related conditions, including situs inversus and polycystic kidney disease. The manuscript is well written but with issues in figure presentation and concerns in some of the experiments. Overall, the findings are novel, and they shed light on the role of peroxisomal cholesterol transport in the functions of ciliary polycystins, and suggests new avenues for understanding the pathogenesis and potential treatment strategies for ADPKD.

Response: Thank you very much for your high evaluation of our manuscript and valuable comments. We tried to revise the manuscript appropriately, so that the revision meets your requirements.

Major:

1. Some of the figure images appear misleading. For instance, in Fig. 1A-C, the text states: "ciliary accumulation of both PC1 and PC2 was inhibited (62% of the level of untreated wild-type cells for PC1, and 50% for PC2)." However, the representative images in Fig. 1A suggest a much more dramatic reduction in PC1 and PC2 within the cilia. A similar discrepancy is observed in other figures, such as Fig. 1D-E. According to Fig. 1E, the average levels of the Filipin III signal in M β CD-treated and Pex14 $^{-/-}$ cells are about 50% of those in untreated or WT cells. Yet, in Fig. 1D, the Filipin III signals appear almost completely absent in these cells.

Additionally, in Figs. 3A and 4A, fluorescence signals (including background noise) seem entirely lost, contrasting sharply with the significant residual signals depicted in Figs. 3B-C and 4B-C. This inconsistency between quantitative data and visual representations raises concerns about the accuracy of the representative images and their alignment with the reported measurements.

Response: Based on your important comments, we changed the images shown in Fig. 1A,B,D, 2C, 3A, and 4B to ones correctly depicting the mean value of the signal intensity of PC2 and Filipin III.

2. What is the mechanism underlying the reduction of PC1 and PC2 levels in cilia following a 1.5-hour treatment with methyl- β -cyclodextrin (M β CD)? While the treatment may block the trafficking of new PC1 and PC2 into cilia, it remains unclear what happens to the polycystins already present in the cilia before treatment. Are these pre-existing proteins internalized through endocytosis? If so, it could be valuable to perform live-cell imaging (using fluorescently tagged polycystins) to observe the internalization process during M β CD treatment. This approach would provide direct evidence of whether endocytic pathways are involved in the ciliary clearance of these proteins and offer insights into their fate under cholesterol-depleting conditions.

Response: Thank you very much for this comment. We are also really interested in the mechanism of PC2 trafficking in and out of the primary cilia. As you suggested, we examined the live-cell imaging of polycystins to observe the internalization process during M β CD treatment. To clarify whether endocytosis is involved in the internalization of pre-existing PC2, we transiently expressed mPkd2-AcGFP and DsRed2-PACT (for observation of the basal body) in mIMCD3 cells. Live imaging for AcGFP showed that the ciliary PC2-AcGFP fusion protein decreased while we have not observed any remarkable endocytosis-like vesicles around the basal body with this PC2 reduction (new Fig. S1), implying that pre-existing ciliary PC2 proteins are internalized through lateral diffusion. We mentioned this point in the revised manuscript (line 134, page 7) as follows:

“Live cell imaging for mPkd2-*Aequorea coerulea* green fluorescent protein (AcGFP) transiently expressed in *Pkd2*-knockout mIMCD3 cells showed that M β CD treatment reduced the ciliary signal level of PC2-AcGFP fusion protein without any remarkable endocytosis-like PC2 vesicles around the basal body (Pedersen LB et al, 2016) (Fig. S1), implying that pre-existing ciliary PC2 proteins are internalized through lateral diffusion.”.

3. The current study uses the ratio of lumen area to spheroid area in 3D-cultured mIMCD3 cells as a readout for the effects of cholesterol depletion. However, has any prior research demonstrated that lumen size in this type of experiment effectively models kidney cyst formation? Additionally, what is the relationship between lumen size and the function of ciliary polycystins in these spheroids.

These questions raise important points. If lumen size is intended to serve as a proxy for *in vivo* cyst formation, it would be helpful to reference studies that have validated this model. If no direct links between lumen size and kidney cyst formation have been established, the study should clarify the rationale for using this readout and discuss its relevance to *in vivo* cyst development.

Response: Thank you very much for this comment. The spheroid technique using IMCD3 cells has been a simple *in vitro* method of studying *in vivo* tissue organization (Giles et al., 3D spheroid model of mIMCD3 cells for studying ciliopathies and renal epithelial disorders, *Nat Protoc*, 2014). This technique has led to the successful identification of potentially pathogenic ciliopathy variants, and it has elucidated pathogenesis. Luijten et al. (Birt-Hogg-Dube syndrome is a novel ciliopathy, *Hum Mol Genet*, 2013) demonstrated that the 3D spheroid of mIMCD3 cells is the established *in vitro* model of *in vivo* kidney cyst formation for renal morphogenesis mimicking renal tubules by analyzing the spheroid and lumen size. In the present study, based on prior research, our findings with *in vitro* 3D spheroid model and *in vivo* mice kidney represent the reverse genetics proof that defects in the interaction between cholesterol and PC2 cause polycystic kidney diseases. We added the following sentence (line 158, page 8) in the Results section: “It was reported that the lumen sizes of ciliopathy-related gene-disrupted mIMCD3 spheroids were enlarged (Giles RH et al, 2014, Luijten MN et al, 2013), suggesting that the ratio of lumen size to spheroid size serves as a readout of primary cilia function in 3D-epithelial architecture.”.

4. The current study assessed PC2 channel currents by expressing PC2 and its mutants in HEK293 cells. However, previous research has shown that in these cells, PC2 predominantly localizes to the ER membrane rather than the plasma membrane, making its channel activity difficult to record. Given the small recorded currents (Fig. 2B), the authors need to provide stronger evidence that these currents indeed originate from the PC2 channel.

To address this, the following points should be clarified:

- Localization of PC2: Do WT and mutant PC2 traffick to the plasma membrane in HEK293 cells? Immunofluorescence or surface biotinylation assays could help confirm this.

- *Baseline controls: What do the currents look like in untransfected or empty vector-transfected cells? These controls are essential to rule out background or nonspecific currents.*
- *Pathogenic mutations: Testing other well-characterized pathogenic mutations like D511V, which abolishes PC2 channel function, would provide further validation.*
- *W414G mutation: The effects of W414G on PC2 channel function remain controversial, as shown in studies by Cai et al. (Altered trafficking and stability of polycystins underlie polycystic kidney disease, JCI, 2014) and Wang et al. (The diverse effects of pathogenic point mutations on ion channel activity of a gain-of-function polycystin-2, JBC, 2023). Discussing these differences would enhance the rigor of the findings.*

Response: Thank you very much for your constructive comment. As you pointed out, we also think that it is desirable to examine the evidence that currents we demonstrated originate from the PC2 channel. According to your comments and another reviewer's suggestions, we measured whole membrane currents in PC2 R325Q mutated cells as a pathogenic mutation and untransfected parent cells as a baseline control (Fig. 2B). The PC2 R325Q mutant, which was reported that the ion channel activity is impaired (Vien TN et al, 2020), stably expressing HEK293 cell line was generated in this revised manuscript. Like the other PC2s examined in this study, PC2 R325Q mutant was visible in the plasma membrane (new Fig. S3B). Overall membrane currents were statistically higher in wild-type PC2-expressing and PC2 L517R-expressing cells compared with the level in PC2 R325Q-expressing ones and untransfected parent ones. These data suggest the surface expression of *Pkd2* on the plasma membrane. The expression level of *Pkd2* in our stably expressing HEK293 cells might be higher comparing to those in previous reports, which may permit slight translocation of channels to the plasma membrane.

On the other hand, your comment about W414G mutation is understandable. Although the effects of W414G on PC2 channel function remain controversial, we think that W414G mutant possesses the channel activity based on the following results. First, the current trace of PC2 W414G-expressing cells was relatively close to that of PC2 wild type-expressing ones rather than that of PC2 R325Q-expressing ones (Fig. 2B). Second, the rescue experiments with cholesterol supplementation to the PC2 W414G-expressing spheroids support the channel function of W414G mutant by restoring lumen-to-spheroid ratio when cholesterol levels are replenished. However, in practice, the results showed that the channel activity of the W414G mutant was not statistically significant compared with either wild type PC2 or R325Q mutant, so we have not yet obtained the definite conclusion. Thus, we simply described that "The ion channel activity of PC2 W414G mutant with the cholesterol-binding ability (Fig. 2A), which remains controversial (Cai Y et al, 2014, Wang Y et al, 2023), is likely no apparent differences

between the wild-type and the L517R mutant (Fig. 2B).” in the Result section (line 192, page 9), without a further discussion about the channel activity of W414G.

5. *What happens to ciliary PC1 in the PC2-L515R and PC2-W412G mutant cells?*

Response: We now provide additional immunofluorescence data obtained in PC2-L515R and PC2-W412G mutant cells that also show a reduction of PC1 localization in the primary cilia (new Fig. S5E,F).

Minor:

1. *Line 83, the three citations only show the interaction between PC1 and PC2, for key studies showing the 1:3 stoichiometry of the PC1/PC2 complex, two other references should be cited:*

1) *Yu et al., Structural and molecular basis of the assembly of the TRPP2/PKD1 complex, PNAS, 2009*

2) *Su et al., Structure of the human PKD1-PKD2 complex, Science, 2018*

Response: Thank you very much for this comment. As you suggested, both published works were cited in the revised manuscript's Introduction (Line 88, page 5).

2. *The authors observed that the reduction in ciliary localization of wild-type PC2 in 233 Tulp3-knockout cells remained unchanged even with sufficient exogenous cholesterol. From this, they concluded that "Tulp3 might be epistatically upstream of cholesterol-dependent ciliary PC2 localization." However, an alternative interpretation could be that both Tulp3 and cholesterol are simultaneously necessary for PC2's trafficking to the cilium. This possibility suggests that these two factors might work in tandem rather than sequentially, with each playing a critical, non-redundant role in facilitating PC2's proper localization.*

Response: Thank you for your reasonable comment. We now amended the description in line 249 on page 11, as below. “The reduction of ciliary localization of wild-type PC2 in Tulp3-knockout cells was not altered even in the presence of sufficient exogenous cholesterol, suggesting that Tulp3 might be epistatically upstream or parallel position of cholesterol-dependent ciliary PC2 localization (Fig. 4B–D)”

3. *Lines 257-259: "implying that the low abundance of ciliary PC2 due to the L515R mutation causes the pathogenesis of ADPKD, even in the presence of sufficient ciliary cholesterol." The authors might consider softening this interpretation. A revised version could read: "implying*

that the low abundance of ciliary PC2 caused by the L515R mutation might contribute to the pathogenesis of this mutation in ADPKD, even in the presence of sufficient ciliary cholesterol." This adjustment leaves room for other potential contributing factors and acknowledges the complexity of the disease mechanism.

Response: Thank you very much for this comment. We have changed the text as suggested (line 274, page 12).

Reviewer #2:

The current study examines cholesterol's role in polycystin-2 (PC2) localization to primary cilia and its implications for polycystic kidney disease (PKD). The authors demonstrate that peroxisome-derived cholesterol is essential for proper PC2 localization in the ciliary membrane, and disruption of this interaction leads to an increased cystic index. They identify a critical cholesterol-binding site on PC2 (L517) and claim that mutation of this site (L517R) impairs ciliary localization without affecting channel function. Using both cell culture models and mouse genetics, they show that proper cholesterol-PC2 interaction is necessary to prevent cyst formation. This work provides mechanistic and therapeutic insights into PKD pathogenesis.

Response: Thank you very much for your high evaluation of our manuscript and valuable comments. We hope that we could answer your questions, so that the revision meets your requirements.

Evaluation of Main Points:

A. Role of cholesterol in PC2 ciliary localization

The authors establish a clear link between cholesterol levels and PC2's presence in primary cilia. The rescue experiments with cholesterol supplementation strongly support their conclusions by restoring PC2 localization when cholesterol levels are replenished.

Response: We appreciate your positive evaluation of our manuscript.

B. Characterization of PC2 L517R Mutation

Strengths: Biochemical pull-downs and imaging studies provide multiple lines of evidence that support their findings about this mutation's effects.

Concerns: The electrophysiology data in Figure 2B requires several important controls for proper interpretation. Incorporating non-transfected control recordings would help establish baseline currents. Recent work from Ha et al. (2024) indicates that PC2 does not form active channels in HEK cells, and it would be helpful to address how the author's findings relate to their observations.

Given the small current amplitudes (pA/pF range), additional validation would strengthen the conclusions. Did the authors consider using standard TRP channel blockers (gadolinium, ruthenium red) to confirm these are PC2-mediated currents? Additionally, characterization of basic channel properties including reversal potentials and ion selectivity would provide valuable validation of these currents. We need experiments to help distinguish PC2-specific channel activity from background conductances.

Response: Thank you very much for your constructive comment. As you pointed out, we also think that it is desirable to examine the evidence that currents we demonstrated originate from the PC2 channel. According to the reviewer's comment, we initially considered experiments using PC2 blockers. However, we thought that available Pkd2 blockers are not sufficiently selective and suppress a broad range of ion channels, making it challenging to isolate the specific contribution of the ion channels mediating the small membrane current observed in Fig. 2B.

To address this issue, we transfected the PC2 R325Q mutant, a pathogenic mutant known to lack channel activity (Vien TN et al, 2020), and examined its membrane current (Fig. 2B). Additionally, according to the reviewer's comment, we assessed the membrane current of untransfected parental HEK293 cells for comparison. To perform these experiments, we generated a stable HEK293 cell line expressing R325Q mutant. We found that overall membrane currents in cells expressing wild-type and the L517R mutant PCs were significantly higher than those in cells expressing R325Q mutant. Furthermore, the current traces in R325Q-expressing cells were identical to that of untransfected parent cells. We confirmed the presence of PC2 R325Q mutant on the plasma membrane (Fig. S3B). These findings collectively suggest both wild-type and mutant PC2 channels are expressed on the plasma membrane.

However, very small PC2 currents may reflect the limited plasma membrane localization of PC2, consistent with previous reports. The expression level of PC2 in our stably expressing HEK293 cells might be higher comparing to those in previous reports, which may permit slight translocation of PC2 channels to the plasma membrane.

C. In Vivo Validation: The generation and characterization of the PC2 L517R knock-in mouse model significantly validate their cellular findings. The mouse phenotypes, including enlarged Bowman's space, renal cysts, and heterotaxy, effectively demonstrate the physiological relevance of cholesterol-PC2 interaction. The correlation between cellular and organismal phenotypes is convincing.

Response: We appreciate your positive evaluation of our manuscript.

Overall Assessment: Despite the concerns about electrophysiology controls, this study makes a valuable contribution to the field. It establishes a novel mechanistic link between cholesterol metabolism and PKD, potentially opening new therapeutic avenues. With the suggested revisions, particularly to the electrophysiology data, this work merits publication in a

good-impact journal.

Response: We appreciate your high evaluation of our manuscript.

January 9, 2025

RE: Life Science Alliance Manuscript #LSA-2024-03063-TR

Prof. Tatsuo Miyamoto
Yamaguchi University
Department of Molecular and Cellular Physiology, Graduate School of Medicine
Minami-kogushi 1-1-1
Ube, Yamaguchi 755-8505
Japan

Dear Dr. Miyamoto,

Thank you for submitting your revised manuscript entitled "Cholesterol ensures ciliary polycystin-2 localization to prevent polycystic kidney disease". We would be happy to publish your paper in Life Science Alliance pending final revisions necessary to meet our formatting guidelines.

- please be sure that the authorship listing and order is correct
- please add the Twitter handle of your host institute/organization as well as your own or/and one of the authors in our system
- the contributions selected for Hiroshi Itoh and Hideshi Kawakami do not qualify them for authorship. Please either update the contributions in our system and the Author Contributions section of the manuscript or let us know if the authors need to be removed (and added eventually to the acknowledgment section)
- please add callouts for Figure S1A-B; S4A,C,D,F; S6A-E to your main manuscript text

A. FINAL FILES:

B. MANUSCRIPT ORGANIZATION AND FORMATTING:

Sincerely,

January 10, 2025

RE: Life Science Alliance Manuscript #LSA-2024-03063-TRR

Prof. Tatsuo Miyamoto
Yamaguchi University
Department of Molecular and Cellular Physiology, Graduate School of Medicine
Minami-kogushi 1-1-1
Ube, Yamaguchi 755-8505
Japan

Dear Dr. Miyamoto,

Thank you for submitting your Research Article entitled "Cholesterol ensures ciliary polycystin-2 localization to prevent polycystic kidney disease". It is a pleasure to let you know that your manuscript is now accepted for publication in Life Science Alliance. Congratulations on this interesting work.

DISTRIBUTION OF MATERIALS:

Again, congratulations on a very nice paper. I hope you found the review process to be constructive and are pleased with how the manuscript was handled editorially. We look forward to future exciting submissions from your lab.

Sincerely,
